 **eLIFE**

# Structure of a bacterial RNA polymerase holoenzyme open promoter complex

**Brian Bae[1†], Andrey Feklistov[1†], Agnieszka Lass-Napiorkowska[2], Robert Landick[3,4], Seth A Darst[1]***

[1]Laboratory for Molecular Biophysics, The Rockefeller University, New York, United States; [2]Edward A. Doisy Department of Biochemistry and Molecular Biology, Saint Louis University School of Medicine, St Louis, United States; [3]Department of Biochemistry, University of Wisconsin-madison, Madison, United States; [4]Department of Bacteriology, University of Wisconsin-Madison, Madison, United States

**Abstract** Initiation of transcription is a primary means for controlling gene expression. In bacteria, the RNA polymerase (RNAP) holoenzyme binds and unwinds promoter DNA, forming the transcription bubble of the open promoter complex (RPo). We have determined crystal structures, refined to 4.14 Å-resolution, of RPo containing *Thermus aquaticus* RNAP holoenzyme and promoter DNA that includes the full transcription bubble. The structures, combined with biochemical analyses, reveal key features supporting the formation and maintenance of the double-strand/single-strand DNA junction at the upstream edge of the −10 element where bubble formation initiates. The results also reveal RNAP interactions with duplex DNA just upstream of the −10 element and potential protein/DNA interactions that direct the DNA template strand into the RNAP active site. Addition of an RNA primer to yield a 4 base-pair post-translocated RNA:DNA hybrid mimics an initially transcribing complex at the point where steric clash initiates abortive initiation and $\sigma^A$ dissociation.

*For correspondence: darst@rockefeller.edu

†These authors contributed equally to this work

Competing interests: The authors declare that no competing interests exist.

## Introduction

Transcription initiation is a major control point of gene expression. The initiation process is best understood in the bacterial system (*Saecker et al., 2011*) where the conserved ~400 kD catalytic core of the RNA polymerase (RNAP or E, subunit composition $\alpha_2\beta\beta'\omega$) combines with the promoter-specificity factor $\sigma^A$ to form the holoenzyme ($E\sigma^A$), which locates promoter DNA and unwinds 12–14 base pairs (bps) of the DNA duplex to yield the transcription-competent open promoter complex (RPo). In the presence of nucleotide substrates, RNA synthesis begins with the formation of an initial transcription complex (RP$_{ITC}$). Before transitioning to a stable elongation complex, steric clash between the elongating RNA transcript and elements of σ set up abortive initiation, where the RNAP repeatedly generates and releases short transcripts without dissociating from the promoter (*McClure et al., 1978*; *Murakami et al., 2002a*; *Goldman et al., 2009*). Eventually, the transcript reaches a length of around 17 nt, where σ dissociation and the transition to the stable elongation complex begins (*Nickels et al., 2005*).

The architecture of $E\sigma^A$ recognition of the key −35 and −10 promoter elements was delineated by the structure of *Thermus aquaticus* (*Taq*) $E\sigma^A$ bound to an upstream fork (us-fork) promoter fragment, but the low resolution (6.5 Å) prevented the visualization of molecular details (*Murakami et al., 2002b*). Although high resolution crystal structures defined key, sequence-specific interactions of σ with the −35 element (*Campbell et al., 2002*), the melted −10 element (*Feklistov and Darst, 2011*), as well as with downstream promoter DNA in the context of holoenzyme (*Zhang et al., 2012*), these structures did not contain the full transcription bubble with the upstream double-strand/single-strand (ds/ss) DNA junction at the upstream edge of the −10 element where transcription bubble formation initiates.

**eLife digest** Inside cells, molecules of double-stranded DNA encode the instructions needed to make proteins. To make a protein, the two strands of DNA that make up a gene are separated and one strand acts as a template to make molecules of messenger ribonucleic acid (or mRNA for short). This process is called transcription. The mRNA is then used as a template to assemble the protein. An enzyme called RNA polymerase carries out transcription and is found in all cells ranging from bacteria to humans and other animals.

Bacteria have the simplest form of RNA polymerase and provide an excellent system to study how it controls transcription. It is made up of several proteins that work together to make RNA using DNA as a template. However, it requires the help of another protein called sigma factor to direct it to regions of DNA called promoters, which are just before the start of the gene. When RNA polymerase and the sigma factor interact the resulting group of proteins is known as the RNA polymerase 'holoenzyme'.

Transcription takes place in several stages. To start with, the RNA polymerase holoenzyme locates and binds to promoter DNA. Next, it separates the two strands of DNA and exposes a portion of the template strand. At this point, the DNA and the holoenzyme are said to be in an 'open promoter complex' and the section of promoter DNA that is within it is known as a 'transcription bubble'. However, it is not clear how RNA polymerase holoenzyme interacts with DNA in the open promoter complex.

Bae, Feklistov et al. have now used X-ray crystallography to reveal the three-dimensional structure of the open promoter complex with an entire transcription bubble from a bacterium called *Thermus aquaticus.* The experiments show that there are several important interactions between RNA polymerase holoenzyme and promoter DNA. In particular, the sigma factor inserts into a region of the DNA at the start of the transcription bubble. This rearranges the DNA in a manner that allows the DNA to be exposed and contact the main part of the RNA polymerase. If the holoenzyme fails to contact the DNA in this way, the holoenzyme does not bind properly to the promoter and transcription does not start.

These findings build on previous work to provide a detailed structural framework for understanding how the RNA polymerase holoenzyme and DNA interact to form the open promoter complex. Another study by Bae et al.—which involved some of the same researchers as this study—reveals how another protein called CarD also binds to DNA at the start of the transcription bubble to stabilize the open promoter complex.

Structures of *Escherichia coli* (*Eco*) transcription initiation complexes containing a complete transcription bubble delineated the overall architecture of the full bubble, but the low resolution of the analyses (between 5.5 and 6 Å resolution) prevented a detailed description of protein/DNA interactions (*Zuo and Steitz, 2015*). Here, we determine crystal structures of *Taq* Eσ$^A$ bound to an us-fork promoter fragment, as well as a complete RPo (*Figure 1*, *Figure 1—figure supplement 1*), refined using diffraction data extending to 4.00 and 4.14 Å-resolution, respectively (*Table 1*, *Figure 1—figure supplement 2*), allowing visualization of key features that stabilize the upstream edge of the transcription bubble. The results also reveal functionally relevant holoenzyme interactions with duplex DNA just upstream of the −10 element and potential protein/DNA interactions that direct the DNA template strand (t-strand) into the RNAP active site. Addition of an RNA primer to yield a 4-bp post-translocated RNA:DNA hybrid mimics RP$_{ITC}$ at the point where steric clash initiates abortive initiation and σ$^A$ dissociation (*Murakami et al., 2002a*; *Kulbachinskiy and Mustaev, 2006*).

## Results

### Overall structure of *Taq* RPo

We combined *Taq* EΔ1.1σ$^A$ (Δ1.1σ$^A$: *Taq* σ$^A$ lacking the N-terminal region 1.1, which is dispensable for in vitro transcription. Region 1.1 is not expected to alter protein/DNA interactions in RPo) with us-fork promoter DNA, which contains a ds −35 element and a mostly ss −10 element (*Figure 1—figure supplement 1*). The resulting complex (423 kD) was crystallized and diffraction

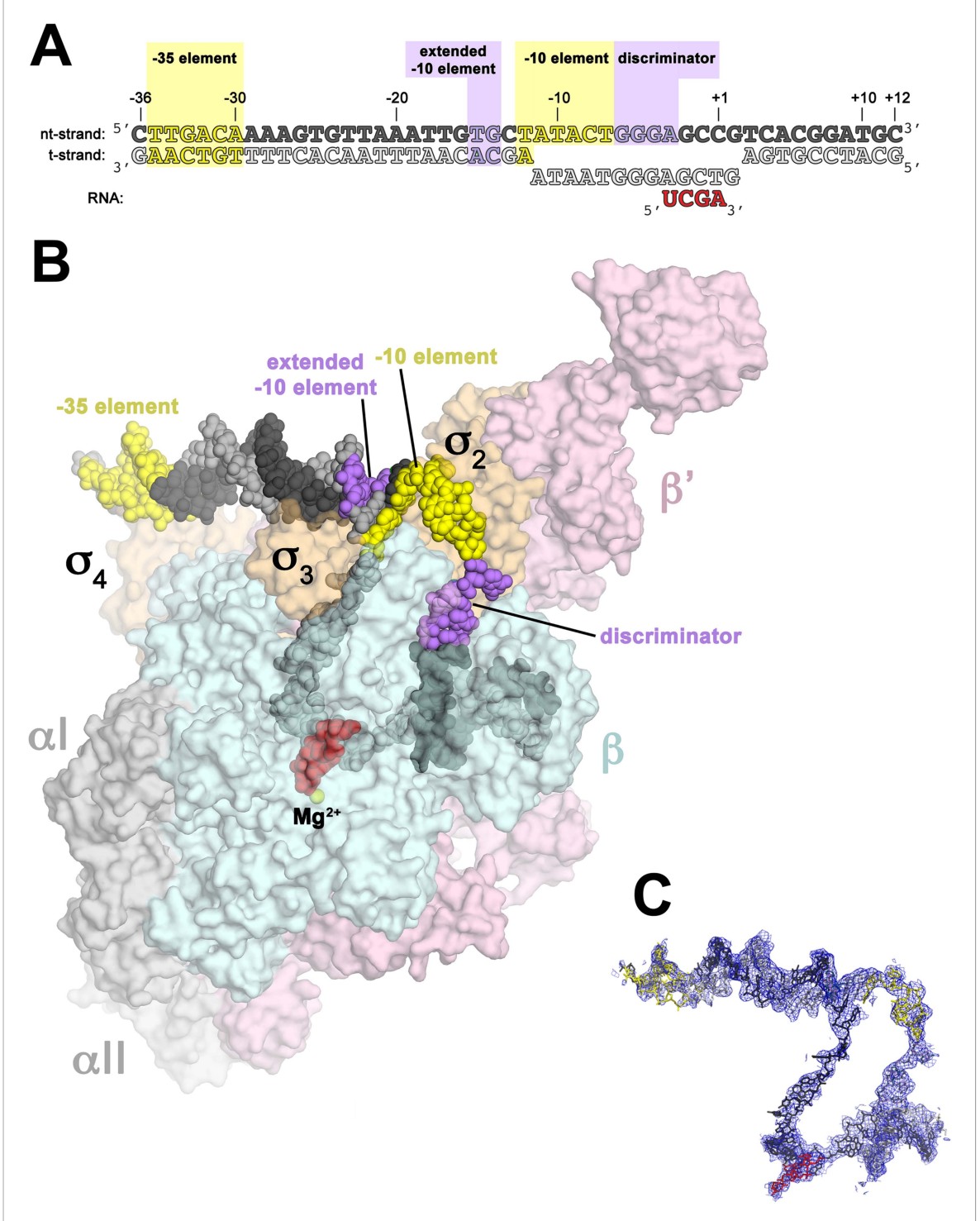

**Figure 1**. Structure of RPo. (**A**) Oligonucleotides used for RPo crystallization. The numbers above denote the DNA position with respect to the transcription start site (+1). The DNA sequence is derived from the full con promoter (*Gaal et al., 2001*). The −35 and −10 (Pribnow box) elements are shaded yellow, the extended −10 (*Keilty and Rosenberg, 1987*) and discriminator (*Feklistov et al., 2006*; *Haugen et al., 2006*) elements purple. The nt-strand DNA (top strand) is colored dark grey; t-strand DNA (bottom strand), light grey; RNA transcript, red. (**B**) Overall structure of RPo. The nucleic acids are shown as CPK spheres and color-coded as above. The *Taq* EΔ1.1σ$^A$ is shown as a molecular surface (αI, αII, ω, grey; β, light cyan; β′, light pink; Δ1.1σ$^A$, light orange), transparent to reveal the RNAP active site Mg$^{2+}$ (yellow sphere) and the nucleic acids held inside the RNAP active site channel. (**C**) Electron density and model for RPo nucleic acids. Blue mesh, $2F_o − F_c$ maps for nucleic acids (contoured at 0.7σ).

*Figure 1. continued on next page*

*Figure 1. Continued*

The following figure supplements are available for figure 1:

**Figure supplement 1**. *(Left)* Synthetic oligonucleotides used for us-fork (−12 bp) crystallization.

**Figure supplement 2**. Data and model quality for us-fork (−12 bp) and RPo complexes.

**Figure supplement 3**. Sequence alignment of regions 2–4 of selected bacterial RNAP primary (Group I) σ subunits.

data were collected and analyzed (*Table 1*). The structure was determined by molecular replacement, which identified two complexes per asymmetric unit, and refined using data extending to 4 Å-resolution (*Table 1*, *Figure 1—figure supplement 2*). The solvent content of the crystals was 82% and examination of the crystal packing revealed space for the expected position of additional promoter DNA. We therefore formed a complete RPo by combining *Taq* EΔ1.1σ$^A$ with a duplex promoter DNA scaffold (−36 to +12 with respect to the transcription start site at +1) but with a non-complementary transcription bubble generated by altering the sequence of the t-strand DNA from −11 to +2. RPo crystallized in the same habit and diffraction data were analyzed to 4.7 Å-resolution (*Table 1*). In the resulting electron density maps, most of the ss t-strand DNA was poorly ordered and unable to be modeled. To stabilize the t-strand DNA, we added an RNA primer complementary to the ss t-strand DNA from +1 to −3, yielding a 4 bp RNA:DNA hybrid (*Figure 1A*). We crystallized the resulting complex (437 kD, which we call RPo hereafter), collected and analyzed diffraction data, and refined the structure using reflections to a minimum Bragg spacing of 4.14 Å (*Table 1*, *Figure 1—figure supplement 2*). In RPo, good electron density for all of the nucleic acids included in the scaffold was observed (*Figure 1C*). The protein/DNA contacts seen in the us-fork complex are essentially identical to the relevant subset of contacts in RPo.

The extensive protein/DNA interface in RPo buries 6300 Å$^2$ of molecular surface (*Figure 1B*). Overall close contacts with the nucleic acids occur from −36 to −30 and −17 to +9, consistent with hydroxyl-radical footprinting of RPo on promoters (*Schickor et al., 1990*; *Ross and Gourse, 2009*). Protein/DNA interactions are absent in the −35/−10 spacer DNA from −29 to −18.

Despite the relatively low resolution of our analysis (*Table 1*), important protein side chain/nucleic acid interactions were resolved in electron density maps. Protein side chain/nucleic acid interactions specifically discussed in this paper are supported by unbiased simulated annealing omit maps shown for each case (see below). The protein side chain/nucleic acid interactions specifically discussed in this paper occur via conserved (often universally) residues of the RNAP β′ or σ$^A$ subunits. The level of conservation of relevant β′ residues, determined from an alignment of 834 bacterial RNAP β′ subunit sequences (*Lane and Darst, 2010*) is tabulated in *Table 2*. An alignment of 1002 diverse σ$^A$ sequences was constructed (Supplementary file 1; a sub-alignment of selected diverse sequences is shown in *Figure 1—figure supplement 3*) and the level of conservation of relevant σ$^A$ residues is tabulated in *Table 3*.

## RNAP interacts with ds DNA just upstream of the −10 element and specifically recognizes the extended −10 element

Starting from the upstream end of the promoter DNA, the −35 element interacts exclusively with $\sigma_4^A$ in a manner consistent with the high-resolution (2.4 Å) structure of the isolated $\sigma_4^A$/−35 element complex (*Campbell et al., 2002*). The duplex DNA just upstream of the −10 element (−17 to −13) interacts with β′, $\sigma_3^A$, and $\sigma_2^A$ (*Figure 1B*).

Previously, conserved residues of the β′-zipper (β′Y34 and, to a lesser extent, β′R35; *Table 2*) that contribute to RPo stability by interacting with duplex spacer DNA were identified (*Yuzenkova et al., 2011*). In the RPo structure, both β′Y34 and β′R35 are positioned to form polar interactions with the −17 nt-strand DNA (−17(nt)) phosphate (*Figure 2A,C*).

We observe many interactions of $\sigma_3^A$ and $\sigma_2^A$ with the duplex DNA just upstream of the transcription bubble (−17 to −12), predominantly with the nt-strand facing the holoenzyme (*Figures 1B, 2*). Conserved H278 and R274 of σ$^A$ (corresponding to *Eco* σ$^{70}$ H455 and R451; *Figure 1—figure supplement 3*; *Table 3*) are positioned to interact with the −17(nt) and −16(nt)

**Table 1.** Table of crystallographic statistics

| Taq EΔ1.1σ$^A$ + | Us-fork (−12 bp) | Us-fork (−11 bp) | Bubble/RNA (RPo) | Bubble |
|---|---|---|---|---|
| **Data collection** | | | | |
| Space group | $P4_32_12$ | $P4_32_12$ | $P4_32_12$ | $P4_32_12$ |
| Combined datasets | 3 | 4 | 10 | 4 |
| Cell dimensions | | | | |
| a (Å) | 289.87 | 288.23 | 289.26 | 290.76 |
| b (Å) | 289.87 | 288.23 | 289.26 | 290.76 |
| c (Å) | 537.36 | 535.25 | 536.60 | 540.84 |
| Wavelength (Å) | 1.075 | 1.075 | 1.075 | 1.075 |
| Resolution (Å) | 50.03–4.01 (4.143–4.01)† | 49.43–4.60 (4.76–4.60)† | 34.96–4.14 (4.29–4.14)† | 40.00–4.74 (4.91–4.74)† |
| Total reflections | 2,192,774 (167,274) | 1,268,008 (123,590) | 5,022,989 (367,167) | 1,849,900 (143,237) |
| Unique reflections | 185,025 (18,323) | 125,012 (11,043) | 172,210 (16,966) | 116,874 (8115) |
| Multiplicity | 11.5 (9.1) | 10.1 (10.1) | 29.2 (21.6) | 15.8 (12.7) |
| Completeness (%) | 99.9 (98.6) | 99.0 (100.00) | 100 (99.8) | 99.6 (97.0) |
| <I>/σI | 6.68 (0.43) | 5.57 (0.60) | 9.4 (0.8) | 8.11 (0.81) |
| Wilson B-factor (Å$^2$) | 133.90 | 154.68 | 101.16 | 196.78 |
| $R_{pim}$‡ | 0.173 (2.136) | 0.238 (1.816) | 0.207 (1.264) | 0.177 (2.047) |
| CC1/2§ | 0.988 (0.219) | 0.975 (0.323) | 0.983 (0.157) | 0.974 (0.205) |
| CC*§ | 0.997 (0.601) | 0.994 (0.698) | 0.996 (0.521) | 0.993 (0.584) |
| **Anisotropic scaling B-factors¶** | | | | |
| a*, b* (Å$^2$) | 18.19 | 22.15 | 15.44 | 20.96 |
| c* (Å$^2$) | −36.37 | −44.3 | −30.88 | −41.92 |
| **Refinement** | | | | |
| $R_{work}$/$R_{free}$ | 0.2531/0.2961 (0.3712/0.4188) | 0.2446/0.2800 (0.3464/0.3726) | 0.270/0.308 (0.358/0.371) | – |
| CC$_{work}$/CC$_{free}$§ | 0.918/0.900 (0.373/0.300) | 0.923/0.904 (0.438/0.293) | 0.897/0.890 (0.343/0.280) | – |
| No. atoms | 56,478 | 56,501 | 58,279 | – |
| Macromolecule | 56,472 | 56,495 | 58,273 | – |
| Ligand/ion | 6 | 6 | 6 | – |
| Water | 0 | 0 | 0 | – |
| Protein residues | 6871 | 6871 | 6875 | – |
| B-Factors | | | | |
| Protein | 139.60 | 175.65 | 137.7 | – |
| Ligand/ion | 169.70 | 175.69 | 134.4 | – |
| R.m.s deviations | | | | |
| Bond lengths (Å) | 0.004 | 0.005 | 0.003 | – |
| Bond angles (°) | 0.91 | 1.12 | 0.80 | – |
| Clashscore | 11.91 | 22.89 | 12.88 | – |
| Ramachandran favored (%) | 94 | 88 | 92 | – |
| Ramachandran outliers (%) | 0.41 | 0.83 | 0.23 | – |

†Values in parentheses are for highest-resolution shell.

‡(**Diederichs and Karplus, 1997**).

§(**Karplus and Diederichs, 2012**).

¶As determined by the UCLA MBI Diffraction Anisotropy Server (http://services.mbi.ucla.edu/anisoscale/).

**Table 2.** Conservation of RNAP β′ subunit residues

| Residue | % Identity* | Blosum62 score*, † | Distribution of residues from alignment* |
|---------|-------------|---------------------|-------------------------------------------|
| β′Y34 | 99.5 | 0.976 | Y 803; H 1; Q 1; F 2 |
| β′R35 | 99.4 | 0.980 | R 829; K 5 |

*Determined from an alignment of 834 bacterial RNAP β′ subunit sequences (**Lane and Darst, 2010**).
†Blosum62 score calculated by PFAAT (**Johnson et al., 2003**).

phosphates, respectively (**Figure 2**). Substitution of either of these residues causes defects in promoter binding (**Barne et al., 1997**; **Fenton et al., 2000**; **Singh et al., 2011**).

Sequence-specific recognition of the duplex DNA upstream of the −10 element can occur through the extended −10 element ($T_{−15}G_{−14}$), which stabilizes RPo and can substitute for the −35 element (**Keilty and Rosenberg, 1987**). Conserved E281 of $\sigma^A_3$ ($\sigma^{70}$ E458; **Figure 1—figure supplement 3**; **Table 3**) is positioned to recognize the −14 GC bp through a polar interaction with $C_{−14}$(t), as predicted from allele-specific suppression genetics (**Barne et al., 1997**) (**Figure 2A,C**). $G_{−14}$(nt) is also specifically recognized by conserved R264 ($\sigma^{70}$ R441; **Figure 1—figure supplement 3**; **Table 3**) of $\sigma^A_2$ (**Figure 2B,D**). Substitutions in the corresponding amino acid position of an alternative σ cause defects in promoter recognition (**Daniels et al., 1990**). Methylation protection and interference indicates *Eco* Eσ$^{70}$ makes close contacts with $G_{−14}$(nt) on an extended −10 promoter (**Minchin and Busby, 1993**). Conserved V277 ($\sigma^{70}$ V454; **Figure 1—figure supplement 3**; **Table 3**) may contact the $T_{−15}$(nt) methyl group, possibly explaining the preference for T at this position (**Figure 2B**).

The primary role of $\sigma_2$ in −10 element recognition was first uncovered when substitutions of invariant Q260 ($\sigma^{70}$ Q437; **Figure 1—figure supplement 3**; **Table 3**) were shown to affect sequence-specific recognition of the −12 bp (**Kenney et al., 1989**; **Waldburger et al., 1990**). Modeling suggested that Q260 may H-bond with the major-groove edge of $A_{−12}$(t) (**Feklistov and Darst, 2011**). However, in our structures, the amide group of the Q260 side chain points away from the major-groove edge of $A_{−12}$(t) and cannot form H-bonds (**Figure 2B,D**). We suggest that Q260 may form base-specific H-bonds with the −12 bp in an intermediate during the pathway to RPo formation (**Saecker et al., 2011**), whereas our structures represent the final, transcription ready RPo, explaining the genetic data.

**Table 3.** Conservation of σ$^A$ residues

| Residue | % Identity* | Blosum62 score*, † | Distribution of residues from alignment* |
|---------|-------------|---------------------|-------------------------------------------|
| σ$^A$ Y217 | 99.4 | 0.988 | 996 Y; 5 H; 1 F |
| σ$^A$ R220 | 100 | 0.998 | – |
| σ$^A$ W256 | 100 | 0.998 | – |
| σ$^A$ W257 | 100 | 0.998 | – |
| σ$^A$ Q260 | 100 | 0.998 | – |
| σ$^A$ R264 | 100 | 0.998 | – |
| σ$^A$ R274 | 100 | 0.998 | – |
| σ$^A$ V277 | 100 | 0.998 | – |
| σ$^A$ H278 | 100 | 0.998 | – |
| σ$^A$ E281 | 100 | 0.998 | – |
| σ$^A$ R288 | 99.7 | 0.993 | 999 R; 3 K |
| σ$^A$ R291 | 99.7 | 0.988 | 997 R; 2 K; 1 H; 1 S; 1 L |

*Determined from an alignment of 1002 bacterial RNAP primary σ subunit sequences (Supplementary file 1).
†Blosum62 score calculated by PFAAT (**Johnson et al., 2003**).

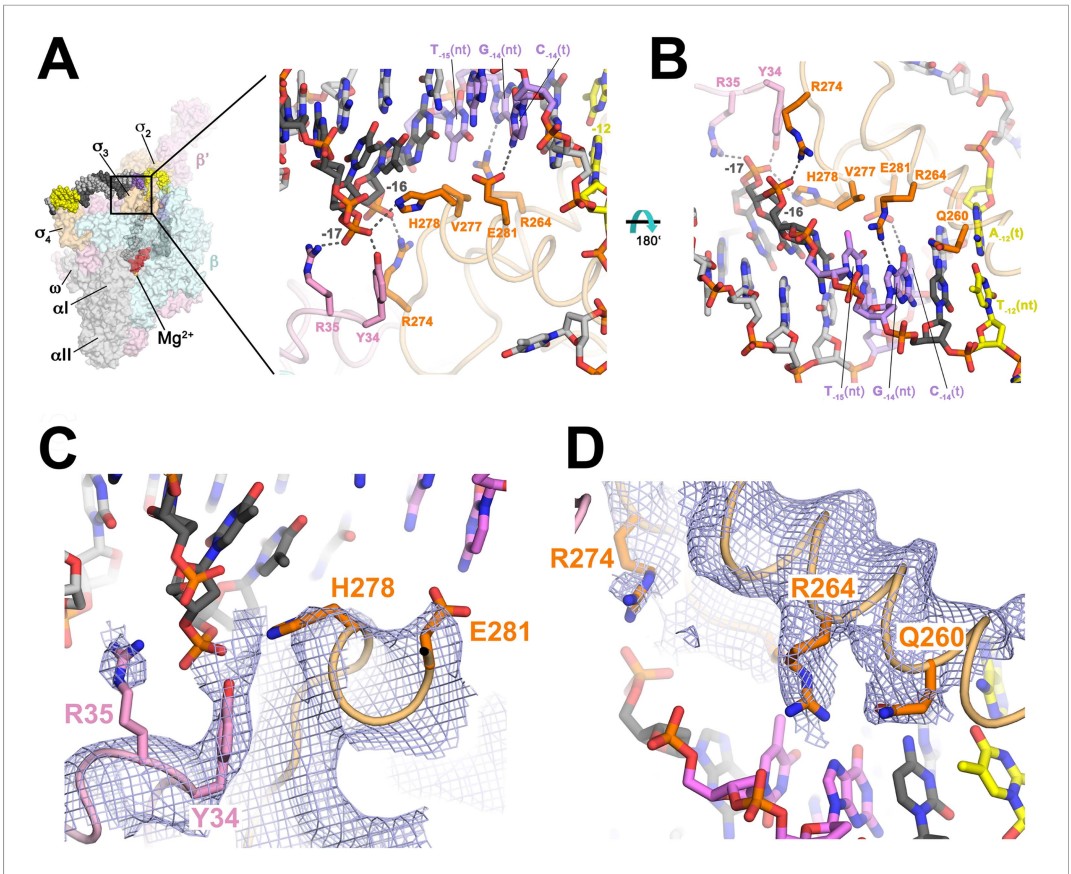

**Figure 2**. Protein interactions with duplex DNA upstream of the transcription bubble and recognition of the extended −10 element. (**A**) (*Left*) Overall view of RPo structure (similar to *Figure 1B*). The boxed area is magnified on the right. (*Right*) Magnified view showing protein interactions (β′ and σ$^A$) with duplex DNA from −18 to −12. Proteins are shown as backbone worms (β′, light pink; σ$^A$, light orange) with interacting side chains shown in stick format (β′, pink; σ$^A$, orange). Likely polar interactions are denoted with grey dashed lines. (**B**) Same as (**A**) (*right*) but rotated 180° about the x-axis. (**C**) Similar view as (**A**) (*right*). Superimposed is the simulated annealing omit map (grey mesh, $2F_o − F_c$, contoured at 1σ), calculated from a model where the following protein segments were removed (β′ 33–36; σ$^A$ 259–292) and shown only within 2 Å of omitted atoms. (**D**) Similar view as (**B**). Superimposed is the simulated annealing omit map (grey mesh, $2F_o − F_c$, contoured at 1σ), calculated from a model where the following protein segments were removed (β′ 33–36; σ$^A$ 259–292) and shown only within 2 Å of omitted atoms. DOI: 10.7554/eLife.08504.010

## Structural role of σ$^A$ aromatic residues in forming and stabilizing the upstream ds/ss junction of the transcription bubble

Flipping of the A$_{−11}$(nt) base from the duplex DNA into its recognition pocket in σ$^A_2$ is thought to be the key event in the initiation of promoter melting (*Chen and Helmann, 1997*; *Lim et al., 2001*; *Heyduk et al., 2006*; *Feklistov and Darst, 2011*). Strand opening propagates downstream to +1, but in the upstream direction, the base-paired T$_{−12}$(nt) interacts with an invariant W-dyad of σ$^A_2$ (W256/W257, σ$^{70}$ W433/W434; *Figure 1—figure supplement 3*; *Table 3*) to maintain the ds/ss (−12/−11) junction at the upstream edge of the transcription bubble (*Figure 3A,C,D*, *Figure 3—figure supplement 1*). The stabilization of the upstream ds/ss junction involves a previously unseen rearrangement of the W256 side chain. In all previous high resolution structures of σ$^A$/σ$^{70}$ in many different contexts but never with an upstream ds/ss junction (*Malhotra et al., 1996*; *Campbell et al., 2002*; *Vassylyev et al., 2002*; *Feklistov and Darst, 2011*; *Zhang et al., 2012*), the W256 side chain makes an 'edge-on' interaction with W257 (*Figure 3B*). In the presence of the upstream ds/ss junction, the W256 side chain rotates away from W257, filling the space vacated by the flipped-out A$_{−11}$(nt) and forming a π-stack with the face of T$_{−12}$(t) otherwise exposed by the absence of A$_{−11}$(nt)

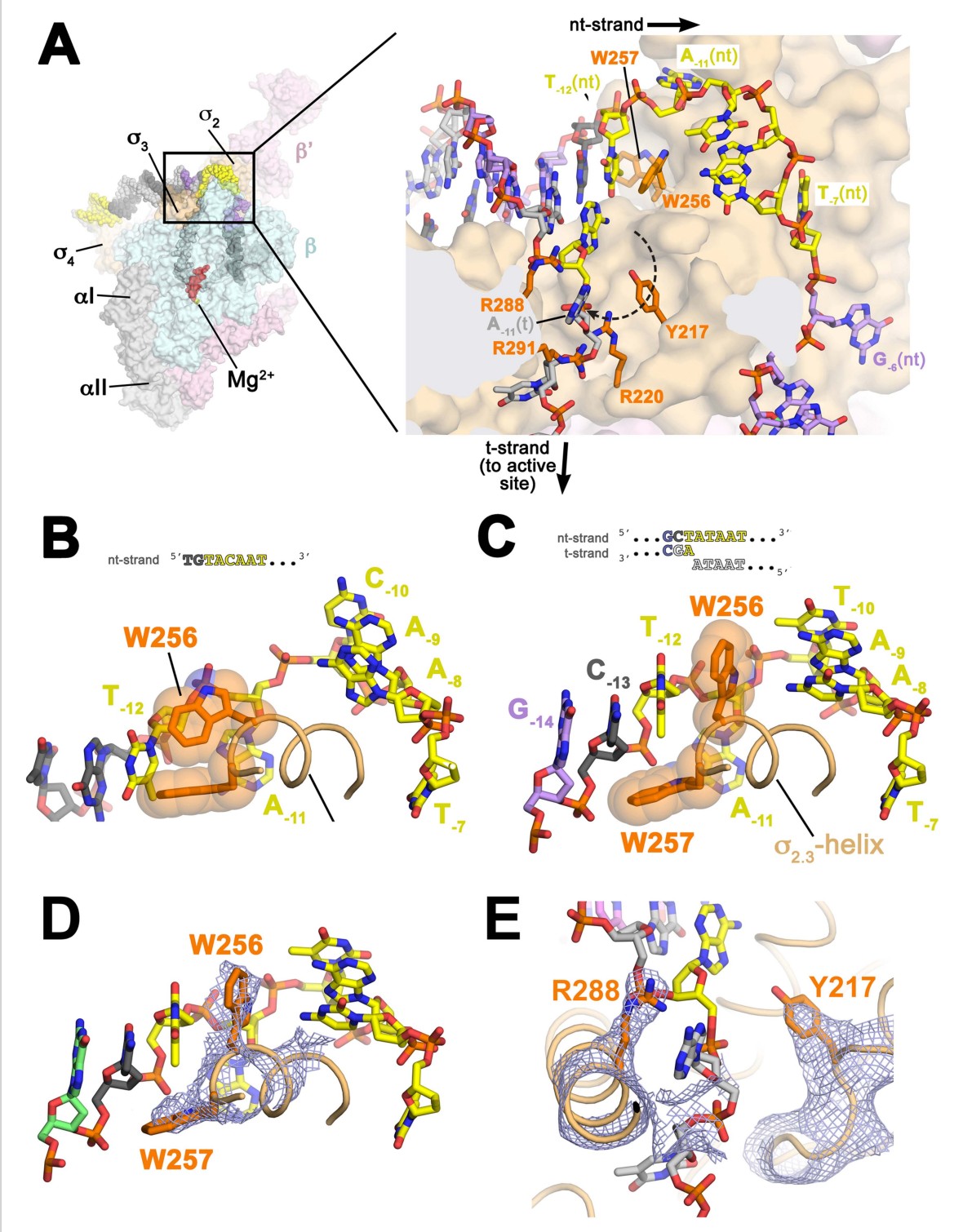

**Figure 3**. Structures maintaining the upstream ds/ss junction of the transcription bubble and directing the t-strand DNA to the RNAP active site. (**A**) (*Left*) Overall view of RPo structure (similar to *Figure 1B*). The boxed area is magnified on the right. (*Right*) Magnified view showing the upstream ds/ss junction of the transcription bubble in RPo (the RNAP β subunit, which obscures the view, has been removed). RNAP is shown as a molecular surface, except side chains of key σ$^A$ residues (R217, R220, W256, R288, R291) are shown (orange). The orthogonal directions of the ss nt- and t-strand DNA following the upstream ds/ss junction are denoted by black arrows. The dashed, curved line denotes the potential path of the t-strand −11 base from its position in the duplex DNA (base-paired to A$_{−11}$(nt)) to its position in the structure. (**B**) Structure of *Taq* σ$_2^A$ bound to the ss, nt-strand −10 element

*Figure 3. continued on next page*

*Figure 3. Continued*

(PDB ID 3UGO) (*Feklistov and Darst, 2011*) showing the disposition of the universally conserved σ$^A$ W-dyad (*Taq* σ$^A$ W256/W257). Shown is the ss DNA from −14 to −7 (−10 element colored yellow), the σ$^A_{2.3}$ − helix (light orange) and the W-dyad (orange side chains with transparent CPK atoms). W256 makes an edge-on interaction with the face of W257, as observed in all other σ$^{70}$/σ$^A$ structures in many different contexts (*Malhotra et al., 1996*; *Campbell et al., 2002*; *Vassylyev et al., 2002*; *Murakami et al., 2002a, 2002b*; *Feklistov and Darst, 2011*; *Zhang et al., 2012*). (**C**) Disposition of the W-dyad in RPo (containing upstream ds/ss junction, shown schematically above). Only the nt-strand DNA from −14 to −7, the σ$^A_{2.3}$ − helix, and the W-dyad are shown (as in **B**). (**D**) Same view as (**C**). Superimposed is the simulated annealing omit map (grey mesh, $2F_o − F_c$, contoured at 1σ), calculated from a model where the following segments of σ$^A$ were completely removed (216–221, 255–258, and 287–292) and shown only within 2 Å of omitted atoms. (**E**) Similar view as (**A**) (*right*). Superimposed is the simulated annealing omit map (grey mesh, $2F_o − F_c$, contoured at 1σ), calculated from a model where the following segments of σ$^A$ were removed (216–221, 255–258, and 287–292) and shown only within 2 Å of omitted atoms. Clear Fourier density for σ$^A$ Y217 and R288 is shown.

The following figure supplement is available for figure 3:

**Figure supplement 1**. Stereo view of RPo model and resulting electron density map (grey mesh, $2F_o − F_c$, contoured at 0.7σ).

(*Figure 3C,D*, *Figure 3—figure supplement 1*). The W-dyad forms a 'chair'-like structure, with W256 serving as the back of the chair, and W257 as the seat, buttressing T$_{−12}$(nt) from the major groove side (*Figure 3A,C,D*). The methyl group of the T$_{−12}$(nt) base approaches the face of the W257 side chain at a nearly orthogonal angle, possibly forming a favorable methyl π interaction (*Umezawa and Nishio, 1998*; *Brandl et al., 2001*) (*Figure 3C*).

Examination of the structure near the upstream ds/ss junction revealed the solvent-exposed aromatic face of a conserved σ$^A_2$ Tyr side chain, Y217 (σ$^{70}$ Y394; *Figure 3A,E*; *Figure 1—figure supplement 3*; *Table 3*), that does not appear to play an important role in the σ structure per se, but lies along the path the −11(t) base could follow from its position in duplex DNA (base-paired to A$_{−11}$(nt)) to its position in the structure when orphaned by the flipped out A$_{−11}$(nt) (dashed line, *Figure 3A*). The −11(t) nucleotide is almost always a T, being complementary to A$_{−11}$(nt), the most highly conserved position of the −10 element (*Shultzaberger et al., 2007*). In the us-fork, the −11(t) nucleotide is absent (*Figure 1—figure supplement 1*), whereas in RPo, the −11(t) nucleotide is an (atypical) A, being part of the engineered non-complementary transcription bubble (*Figure 1A*). In RPo, the A$_{−11}$(t) base is not stacked on Y217 but instead is about 12 Å away, flipped up alongside the σ$^A_3$ − 3.0 α-helix, sitting between R288 and R291 (*Figure 3A*; *Figure 1—figure supplement 3*; *Table 3*). We reasoned that we may not observe the orphaned −11(t) base stacked on Y217 for two reasons that are not mutually exclusive. First, Y217 may play an important role in stabilizing the melted state of the −11 bp during an intermediate of the normal promoter melting pathway (*Saecker et al., 2011*). Second, structural modeling suggested that the A$_{−11}$(t) purine base present in the synthetic promoter construct (*Figure 1A*) may be too bulky to stack on Y217, which sits at the bottom of a narrow trough in the σ$^A_2$ structure (*Figure 3A*).

To investigate the role of Y217 further, we crystallized *Taq* EΔ1.1σ$^A$ with an us-fork template containing a complementary A:T bp at the −11 position (us-fork (−11 bp); *Figure 4A*). To avoid model bias, we determined the structure by molecular replacement using the *Taq* EΔ1.1σ$^A$/us-fork (−12 bp) structure (lacking the −11(t) base; *Figure 1—figure supplement 1*). The structure was modeled and refined (4.6 Å-resolution, *Table 1*, *Figure 4—figure supplement 1*), and the unbiased density maps revealed clear difference density for the T$_{−11}$(t) base stacked on Y217 (*Figure 4B*).

## Functional role of σ$^A$ aromatic residues in forming and stabilizing the upstream ds/ss junction of the transcription bubble

A functional role for W256 in promoter melting was first proposed by *Helmann and Chamberlin (1988)*. Ala substitution of the corresponding Trp in *Bacillus subtilis* σ$^A$ gave rise to severe promoter melting defects in vitro and corresponding cold phenotypes in vivo (*Juang and Helmann, 1994*; *Panaghie et al., 2000*). The functional role of Y217 has not, to our knowledge, been previously examined.

We investigated the effects of individual Ala substitutions in *Eco* σ$^{70}$ W433 and Y394 (*Taq* W256 and Y217) on the kinetics of RPo formation (*Roe et al., 1984*; *Buc and McClure, 1985*) using a recently reported fluorescence assay (*Ko and Heyduk, 2014*). The assay relies on a Cy3 fluorophore attached to the promoter nt-strand at position +2; fluorescence yield in this context is sensitive to the local environment and increases more than twofold upon RPo formation. Unlike previously used

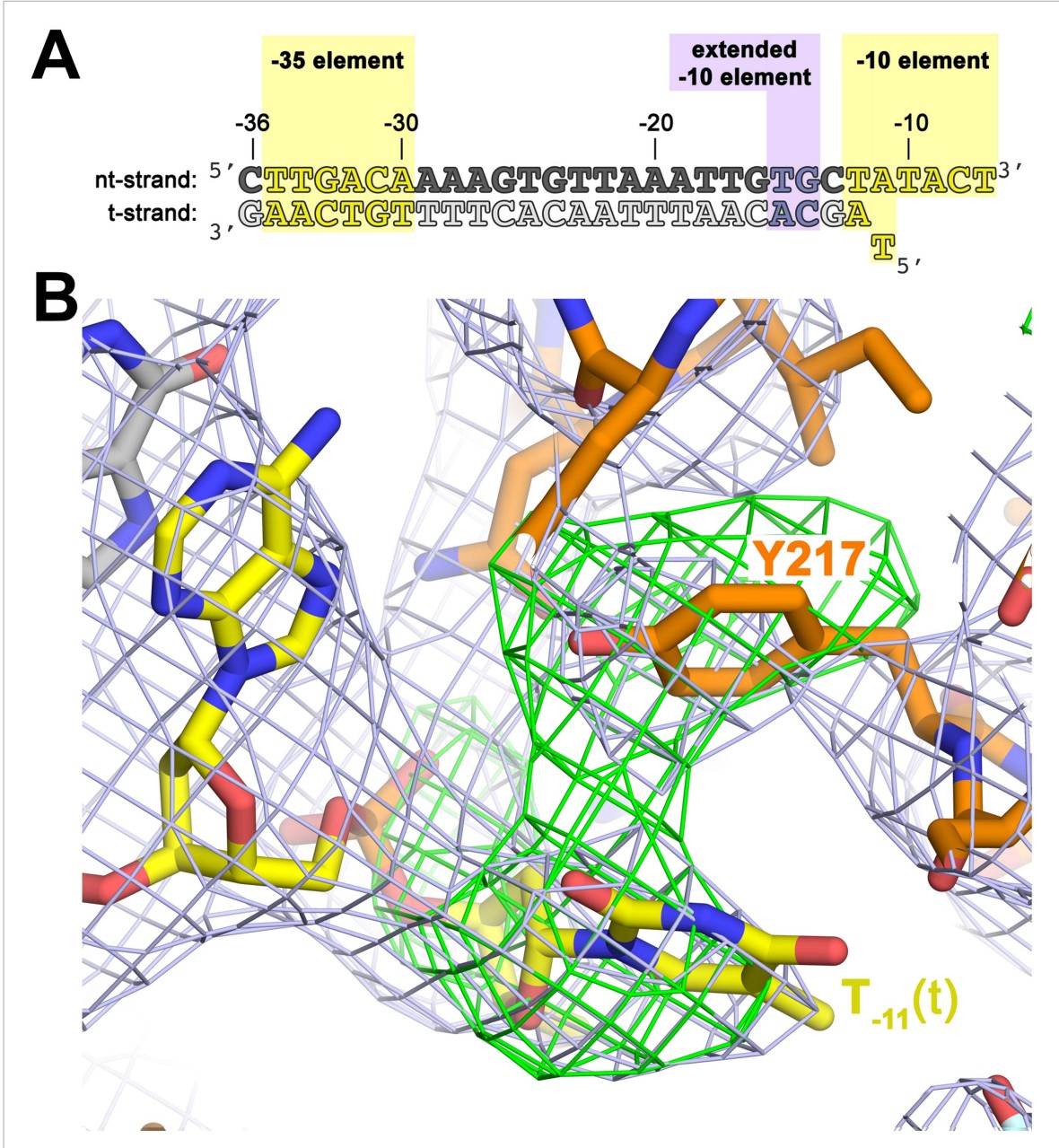

**Figure 4**. The σ$^A$ Y217 may stack on the T$_{-11}$(t) base orphaned by the flipped out A$_{-11}$(nt) base. (**A**) Synthetic oligonucleotides used for us-fork (−11 bp) crystallization. The numbers above the sequence denote the DNA position with respect to the transcription start site (+1). The DNA sequence is derived from the full con promoter (*Gaal et al., 2001*). The −35 and −10 (Pribnow box) elements are shaded yellow, the extended −10 element (*Keilty and Rosenberg, 1987*) purple. The nt-strand DNA (top strand) is colored dark grey; the t-strand DNA (bottom strand), light grey; the RNA transcript, red. (**B**) The T$_{-11}$(t) base orphaned by the flipped out A$_{-11}$(nt) stacks on σ$^A$ Y217 in the us-fork (−11 bp) structure. The 4.6 Å-resolution electron density map (contoured at 0.7σ) is shown (grey mesh). Also superimposed is the simulated annealing omit map (green mesh, $F_o − F_c$, contoured at 3σ), calculated from a model where σ$^A$ Y217 was mutated to Ala and the T$_{-11}$(t) nucleotide was deleted.

The following figure supplement is available for figure 4:

**Figure supplement 1**. Data and model quality for us-fork (−11 bp) complex.

non-equilibrium methods (EMSA, filter binding), this assay allows detection of promoter melting at equilibrium and does not depend on the use of competitors, such as heparin. For these assays, we used one of the most thoroughly characterized promoters, λ P_R (*Saecker et al., 2002, 2011*). Control assays showed that under saturating conditions, both σ70 substitutions (W433A and Y394A) associated with core RNAP and supported abortive transcription as well as wild-type σ70 (data not shown), confirming their structural integrity.

The multistep process of promoter opening can be described by a simplified kinetic scheme (*Figure 5A*) (*McClure, 1980*) where an initial promoter complex (RP_i) existing in rapid equilibrium with free promoter and RNAP (binding step described by a dissociation constant $K_d$) is converted in a rate-limiting step to RPo (isomerization described by the rate constant $k_2$). Fluorescence traces of RPo formation under pseudo first-order conditions (*Roe et al., 1984*) recorded at increasing RNAP concentrations were fit to single-exponentials and yielded observed rate constants ($k_{obs}$) for RPo formation (*Figure 5B*). Nonlinear fits to the resulting hyperbolic curves (*Figure 5C*) allowed the determination of $K_d$ and $k_2$ (*Saecker et al., 2002*) (*Figure 5D*).

Neither σ70 W433A nor Y394A had a significant effect on $K_d$ for RP_i formation, but the substitutions decreased the rate of isomerization by about twofold to threefold (at 37°C, *Figure 5D*). At suboptimal temperature (25°C) the effect of the W433A substitution was more pronounced, resulting in an ~sevenfold reduction in isomerization rate. Neither σ70 W433A nor Y394A significantly altered the affinity of holoenzyme binding to ss oligos comprising the nt-strand of the −10 element (*Tomsic, 2001*) (*Figure 5E*).

W256 appears to make the primary contribution to maintaining the ds/ss junction at the upstream edge of the transcription bubble (*Figure 3A*), suggesting that this residue may play an important role in preventing transcription bubble collapse and dissociation of RPo. To probe the roles of both σ70 W433 and Y394 in maintaining RPo stability, we rapidly destabilized preformed RPo with 1.1 M NaCl (*Gries et al., 2010*) and followed the loss of RPo by monitoring the decay of fluorescence intensity with time (*Figure 5—figure supplement 1*). The dissociation curves are complex, reflecting the detection of a short lived intermediate (expected under these conditions) (*Gries et al., 2010*) by this assay. Although a full analysis is beyond the scope of this study, the overall apparent rate of RPo decay ($k_{off}^{app}$) was determined from single-exponential fits of the decay curves. The σ70 W433A and the Y394A variants both gave a ~fourfold higher rate of RPo dissociation under high salt conditions than did wild-type σ70 (*Figure 5D*, *Figure 5—figure supplement 1*).

## σ^A directs the ss t-strand to the RNAP active site

Downstream from the point of melting, the two DNA strands are directed on orthogonal paths (black arrows, *Figure 3A*). The nt-strand (−11 to −4) drapes across the surface of $\sigma_2^A$, directed by phosphate backbone interactions and notable base-specific recognition of A_{−11}(nt) and T_{−7}(nt) of the −10 element, and G_{−6}(nt) of the discriminator (*Feklistov and Darst, 2011*; *Zhang et al., 2012*). Further downstream, interactions of the nt-strand from −3 to +2 occur exclusively with the RNAP β subunit, including base-specific recognition of G_{+2}(nt) (*Zhang et al., 2012*).

At the point of melting, a ~90° turn of the t-strand backbone (between −12 and −11) may be effected by electrostatic interactions between conserved basic residues of $\sigma_2^A$ (R220; *Figure 1—figure supplement 3*; *Table 3*) and $\sigma_3^A$ (R288, R291) and four t-strand backbone phosphates in a row (−13, −12, −11, −10) encompassing the turn (*Figure 3A*). Strong simulated annealing omit $2F_o − F_c$ density is associated wth $\sigma_3^A$ R288, confirming its role in interacting with the −13(t) phosphate (*Figure 3E*). The $\sigma_2^A$ R220 and $\sigma_3^A$ R291 give weaker difference density so their role in interacting with the −12(t) and −11(t) phosphate groups is tentative. The turn directs the t-strand away from the nt-strand and towards the RNAP active site (*Figure 3A*). The ss t-strand DNA from −9 to −5 is guided towards the RNAP active site through a tunnel formed between the RNAP β1-lobe (called the protrusion in eukaryotic RNAP II; *Cramer et al., 2001*) and the $\sigma_{3.2}$-loop (also referred to as the σ-finger), an extended linker that loops into and out of the RNAP active-site channel (*Murakami et al., 2002a*; *Zhang et al., 2012*), connecting the $\sigma_3$ and $\sigma_4$ domains (*Figure 6*).

## The $\sigma_{3.2}$-loop sterically blocks extension of the 4 nt RNA transcript

Previous structural analyses predicted that the $\sigma_{3.2}$-loop would physically occupy the path of the elongating RNA and must be displaced for full RNA extension to occur (*Vassylyev et al., 2002*; *Murakami et al., 2002a*). Indeed, the upstream edge of the post-translocated 4-nt transcript fits

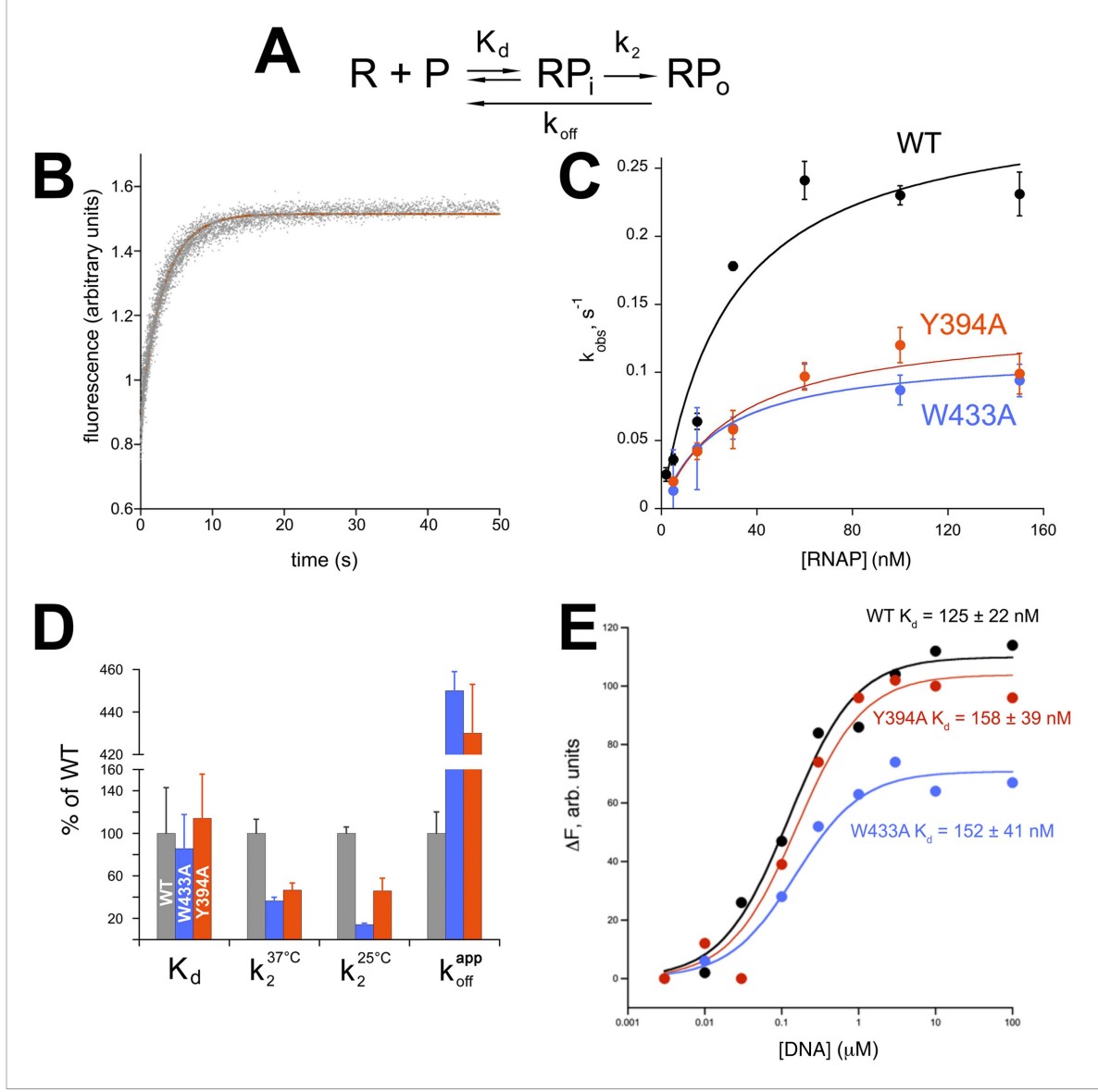

**Figure 5**. Functional role of *Eco* σ[70] W433 and Y394 in RPo formation. (**A**) Simplified, two-step kinetic scheme for RPo formation (*Roe et al., 1984*; *Buc and McClure, 1985*) (R, RNAP; P, promoter; $RP_i$, intermediate complex). (**B**) Representative time trace of fluorescence increase (from Cy3 labelled promoter DNA) during RPo formation. The solid red line illustrates the non-linear regression fit to a single-exponential model (see 'Materials and methods'), which described >90% of the fluorescence amplitude rise. (**C**) The RNAP-concentration dependence of the observed rate ($k_{obs}$) of RPo formation detected by Cy3 fluorescence (*Ko and Heyduk, 2014*) for *Eco* holoenzymes with σ[70] (wt) as well as σ[70] carrying substitutions W433A or Y394A. Error bars denote standard errors of the mean for ≥three independent measurements. (**D**) Summary of effects of σ[70] W433A and Y394A substitutions on thermodynamic and kinetic parameters of RPo formation. The data was normalized to the % observed with wild-type Eσ[70]. (**E**) Equilibrium binding of ss nt-strand oligos of λ $P_R$ promoter −10 element detected in the fluorescent RNAP beacon assay (*Feklistov and Darst, 2011*; *Mekler et al., 2011*) to *Eco* holoenzymes with σ[70], as well as σ[70] carrying substitutions W433A or Y394A.

The following figure supplement is available for figure 5:

**Figure supplement 1**. RPo dissociation data.

snugly between the RNAP active site and the distal tip of the σ[3.2]-loop, which contacts the upstream RNA:DNA bp at −3, and the t-strand bases at −4 and −5 (*Figure 6*). Extension of the RNA transcript and translocation to form a 5 bp post-translocated RNA:DNA hybrid cannot occur without displacement of the σ[3.2]-loop (*Basu et al., 2014*), marking the point in transcription initiation

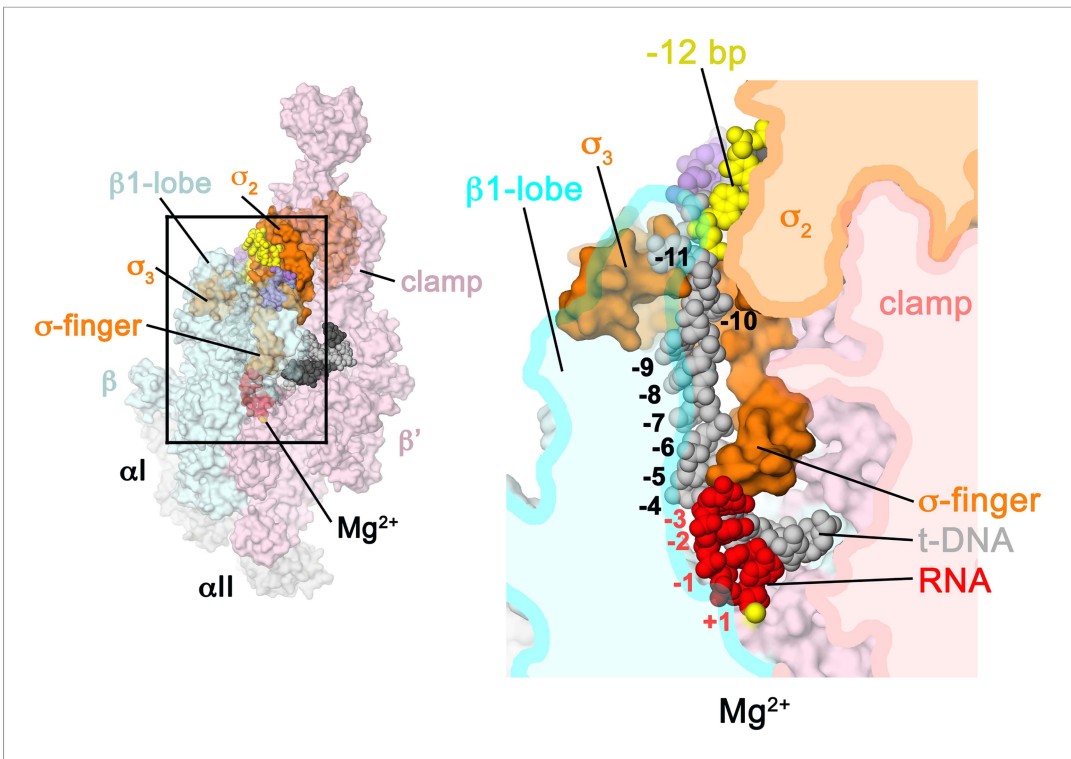

**Figure 6**. Structural role of the $\sigma_{3.2}$-loop. (*Left*) Overall view of RPo structure, colored as in *Figure 1* except $\sigma^A$ is orange. The RNAP β and β′ subunits are transparent to reveal the RNAP active site $Mg^{2+}$ (yellow sphere) and the nucleic acids held inside the RNAP active site channel. The ss nt-strand DNA is omitted for clarity. The boxed area is magnified on the right. (*Right*) Magnified view showing a cross-section of the RNAP active site channel. For clarity, the RNAP β, β′, and $\sigma_2^A$ domains are shown mostly as outlined shapes, with β transparent. The ss t-strand DNA (−11 to −4) is directed towards the RNAP active site through a tunnel between the $\sigma_{3.2}$-loop and the β1-lobe. The 4-nt RNA transcript (−3 to +1) contacts the distal tip of the $\sigma_{3.2}$-loop. Further elongation of the RNA would require displacement of the $\sigma_{3.2}$-loop.

(translocation of the 4–5 bp RNA:DNA hybrid from pre- to post-translocated) where steric clash between the elongating RNA transcript and the $\sigma_{3.2}$-loop begins effecting abortive initiation and σ release (*Murakami et al., 2002a*; *Nickels et al., 2005*; *Kulbachinskiy and Mustaev, 2006*).

## Discussion

Our structures reveal that the overall architecture of the *Taq* RPo (*Figure 1*) closely resembles that of the *Eco* RPo (*Zuo and Steitz, 2015*), but the improved resolution of our analysis allows a more detailed description of protein/DNA interactions (*Figure 2*), particularly interactions involved in forming and stabilizing the ds/ss junction at the upstream edge of the transcription bubble (*Figure 3*). Previous models of RPo were pieced together from structures of σ domains or RNAP holoenzyme complexed with promoter fragments (*Campbell et al., 2002*; *Murakami et al., 2002b*; *Feklistov and Darst, 2011*; *Zhang et al., 2012*). The *Taq* RPo structure upstream of the −10 element matches the overall architecture of the low-resolution (6.5 Å) *Taq* RNAP holoenzyme/upstream-fork promoter complex (*Murakami et al., 2002b*) except unlike the upstream-fork structure (where the RNAP holoenzyme/−35 element interactions were distorted by crystal packing interactions), the *Taq* RPo recapitulates the $\sigma_4$/−35 element interactions seen in the high-resolution (2.4 Å) crystal structure of the *Taq* $\sigma_4^A$/−35 element DNA complex (*Campbell et al., 2002*). The *Taq* RPo structure also recapitulates the $\sigma_2$/−10 element interactions seen in high-resolution (2.1 Å) structures of *Taq* $\sigma_2^A$ complexes with ss −10 element DNA (*Feklistov and Darst, 2011*). The interactions of the RNAP holoenzyme with the ss discriminator element (ss nt-strand DNA from −6 to −3; *Figure 1A*), the ss nt-strand DNA from −2 to +2 (including base-specific interactions of $G_{+2}$(nt) with a pocket in the

RNAP β subunit), and the downstream edge of the transcription bubble and downstream duplex DNA are very similar to those observed in a 2.9 Å-resolution structure of *Tth* RNAP holoenzyme complexed with a downstream-fork promoter template (*Zhang et al., 2012*).

## Role of conserved σ$^A$ aromatic residues in promoter opening

Our results clarify the role of the universally conserved W-dyad of housekeeping (also called primary or group 1) σ's (*Gruber and Bryant, 1997*) in the promoter opening pathway, particularly for *Taq* σ$^A$ W256 (*Eco* σ$^{70}$ W433), which rotates into the DNA duplex and serves as a steric mimic of the flipped-out A$_{-11}$(nt) base by a stacking mechanism (*Figure 3A,C,D*). The bacterial RNAP σ subunit can be added to the list of proteins using a wedge residue (usually an aromatic side chain) to invade the DNA duplex to stabilize the extrahelical conformation of a flipped-out base (*Lau et al., 1998*; *Davies et al., 2000*; *Yang et al., 2009*; *Yi et al., 2012*). We also identified another conserved σ$^A$ aromatic residue (*Taq* σ$^A$ Y217) that plays an important role in the promoter opening pathway, possibly by stacking with T$_{-11}$(t) orphaned when the conserved A$_{-11}$(nt) base flips out (*Figure 4B*).

The kinetic studies reveal that both aromatic residues (W256 and Y217) act in a context dependent manner—they are not important for the initial promoter binding step (*Figure 5D*) nor for binding the ss −10 element DNA (*Figure 5E*): instead W256 and Y217 act to increase the rate of the isomerization (promoter opening step) itself (*Figure 5E,D*), possibly by making contacts unique to the transition state that lower the energy barrier between RPi and RPo in the two-step kinetic scheme (*Figure 5A*). Since the initial promoter binding step (formation of RPi, *Figure 5A*) is not affected by the σ$^{70}$ W433A substitution (*Figure 5D*), we surmise that RPi does not feature the stacking interaction formed by W433A on the T$_{-12}$(nt) base (exposed by the flipping-out of A$_{-11}$(nt)). Since the −11 bp is thought to be the first bp disrupted in the promoter opening pathway (*Chen and Helmann, 1997*; *Lim et al., 2001*; *Heyduk et al., 2006*; *Feklistov and Darst, 2011*), this implies that RPi is a closed complex (RPc) comprising duplex promoter DNA.

The effects of σ$^{70}$ W433A that we observed are consistent with previous observations using nonequilibrium methods (*Fenton et al., 2000*; *Tomsic, 2001*; *Fenton and Gralla, 2003*; *Schroeder et al., 2009*). These observations support the critical role of σ$^A$ W256 and Y217 (σ$^{70}$ W433 and Y394) in formation and stability of RPo.

In addition to the housekeeping σ (σ$^A$ in *Taq* or σ$^{70}$ in *Eco*) that controls transcription of the majority of cellular genes (with consensus −35 and −10 elements of TTGACA and TATAAT, respectively; *Shultzaberger et al., 2007*), bacteria rely on alternative σ's to direct RNAP to highly specialized promoters (with alternative −35 and −10 elements) controlling operons in response to environmental and physiological cues (*Gruber and Gross, 2003*; *Feklistov et al., 2014*). Although the W-dyad is universally conserved in housekeeping σ's (*Gruber and Bryant, 1997*), it is not a conserved feature of alternative σ's (*Lonetto et al., 1992*; *Helmann, 2002*; *Campbell et al., 2003*); bulky hydrophobic residues are favored at the corresponding positions of alternative σ's (but rarely W). The W-dyad is likely to be the optimal configuration for supporting the upstream ds/ss junction of the transcription bubble, giving the housekeeping σ's a powerful DNA-melting capacity, allowing them to function on thousands of highly divergent, nonoptimal promoter sequences. Alternative residues supporting the upstream ds/ss junction of the transcription bubble may weaken the ability of RNAP with alternative σ's to form RPo, fine-tuning their specificity (*Feklistov et al., 2014*). The residue corresponding to *Taq* σ$^A$ Y217 (σ$^{70}$ Y394) appears to be conserved as either Y or F among σ$^{70}$-family alternative σ's suggesting that this residue plays a key role common to all σ's.

## Transcript elongation, scrunching, and σ-release

*Zuo and Steitz (2015)* soaked crystals of *Eco* transcription initiation complexes (containing a full transcription bubble) with NTP substrates to generate short transcripts (with 5′-triphosphate) *in crystallo*. A pre-translocated 4-nt transcript did not reach the σ$_{3.2}$-loop, whereas a pre-translocated 5-nt transcript appeared to just reach and interact with the σ$_{3.2}$-loop. Attempts to generate longer transcripts resulted in severe degradation of the crystals, suggesting significant conformational changes of the RNAP that were incompatible with the crystal packing either due to transcript/σ$_{3.2}$-loop interactions, 'scrunching' of the t-strand DNA (*Kapanidis et al., 2006*; *Revyakin et al., 2006*; *Roberts, 2006*), or both. The upstream edge of our post-translocated 4-nt transcript is equivalent to the pre-translocated 5-nt transcript observed by *Zuo and Steitz (2015)*: in both cases the upstream edge of the RNA just contacts the σ$_{3.2}$-loop and the conformation of

the σ$_{3.2}$-loop is very similar indicating that, at least in this case, the presence or absence of the 5′-triphosphate does not alter the gross interaction of the elongation transcript with the σ$_{3.2}$-loop. In vitro, RNAP initiates efficiently with dinucleotide primers lacking a 5′-triphosphate without obvious defects in σ release or promoter escape.

Basu et al. (2014) were able to generate a 6-nt pre-translocated transcript (containing a 5′-triphosphate) in crystals of *Tth* transcription initiation complexes with a downstream-fork promoter template that lacks duplex DNA upstream of the −10 element and is therefore unable to 'scrunch' the t-strand DNA. In this case, the 5′-nt of the transcript displaces the σ$_{3.2}$-loop, which is not modeled and presumably disordered. Other conformational changes of the RNAP or changes in σ/RNAP interactions were not observed.

### Relationship to RPo formation in eukaryotes

In vitro, the rate-limiting step of bacterial RNAP transcription is often the isomerization step to open the promoter and form RPo (*McClure, 1980*, *1985*; *Amouyal and Buc, 1987*). The kinetics of the many steps of the transcription cycle in vivo have not been characterized, but many transcription units are clearly controlled at the initiation step (*Paul et al., 2004*). In bacteria, recognition of the promoter −10 element and DNA opening are directly coupled (*Feklistov and Darst, 2011*; *Liu et al., 2011*), with the Trp stacking interaction (*Figure 3A,C*) playing a key role.

In contrast to tight coupling between promoter recognition and transcription bubble formation at most bacterial promoters, in eukaryotes promoter recognition, RNAP II recruitment, and promoter opening appear to be uncoupled. The preinitiation complex (PIC) is the molecular assembly through which eukaryotic RNAP II locates and utilizes a promoter, which may be pre-recognized by basal transcription factors. RPo formation requires ATP hydrolysis by the Ssl2 (XPB) subunit of TFIIH, which translocates downstream DNA into RNAP II against fixed upstream contacts to force DNA melting (*Kim et al., 2000*; *Grünberg and Hahn, 2013*). This contrasts with the spontaneous unwinding driven by RNAP/promoter DNA interactions alone during bacterial RPo formation (*Liu et al., 2011*).

Although there are clear similarities between σ and the eukaryotic basal transcription factor IIB in the contacts made to the 5′ RNA, hybrid junction, and ss-tDNA, there is no structural similarity between σ and TFIIB (*Kostrewa et al., 2010*; *Liu et al., 2010*; *Sainsbury et al., 2013*). These contacts may play similar roles in aiding promoter escape by helping eject σ or TFIIB from the RNAP active site cleft, but it is currently unclear whether any eukaryotic basal transcription factor stabilizes an upstream fork-junction by interactions similar to the σ-mediated Trp stacking (*Figure 3A,C*). Further, although effects on RPo formation may help regulate some eukaryotic promoters (*Kouzine et al., 2013*), other steps, including removal of nucleosomes and promoter-proximal pausing (*Boeger et al., 2003*; *Adelman and Lis, 2012*) appear to be rate-limiting at many eukaryotic promoters. Even when promoters are nucleosome-free, assembly of the PIC, rather than promoter opening, may be rate-limiting. Further mechanistic and structural studies of RNAPII on promoters with diverse architectures, including both TATA-containing and TATA-less promoters, are needed for a better understanding of the steps in RNAPII initiation.

### Conclusions

The structures of RPo determined here reveal how the RNAP holoenzyme recognizes the extended −10 element, stabilizes the transcription bubble, directs the t-strand DNA into the RNAP active site, and how the RNA:DNA hybrid initiates σ$^A$ release. Supported by the real-time kinetic data, the structures elucidate the roles of individual aromatic amino acid residues in nucleation of the transcription bubble and maintenance of RPo stability, in part through previously unobserved stacking mechanisms. The results also provide a basis for more incisive investigations of RPo formation and transcriptional regulation (*Bae et al., 2015*).

## Materials and methods

### Preparation and crystallization of *Taq* Δ1.1σ$^A$-holoenzyme/promoter complexes

*Taq* core RNAP and Δ1.1σ$^A$ were prepared as described previously (*Murakami et al., 2003*). Promoter DNA strands (Oligos Etc.) were annealed in 10 mM Tris–HCl, pH 8.0, 1 mM EDTA, 0.2 M NaCl and aliquots were stored at −20˚C.

For crystallization, aliquots of purified *Taq* core RNAP and Δ1.1σ$^A$ were thawed on ice and buffer-exchanged into crystallization buffer (20 mM Tris–HCl, pH 8.0, 0.2 M NaCl). *Taq* Δ1.1σ$^A$-holoenzyme was formed by adding 1.2-fold molar excess of Δ1.1σ$^A$ to the core RNAP and the mixture was incubated for 15 min at room temperature. A 1.5-fold molar excess of promoter DNA was then added to the holoenzyme along with MgCl$_2$ (10 mM final) and incubated for 15 min at room temperature. When present, a fivefold molar excess of RNA primer (GE Dharmacon, Lafayette, CO, United States) was also added. The final RNAP concentration was adjusted to 25 μM. Crystals were grown by vapor diffusion at 22°C by mixing 1 μl of sample with 1 μl of reservoir solution (20 mM MgCl$_2$, 20 mM Tris–HCl, pH 8.0, 1.6 M ammonium sulfate) in a 48-well hanging drop tray (Hampton Research, Aliso Viejo, CA, United States). Thin rod-shaped crystals (typically, 30 × 30 × 300 μm) appeared after about 5 days. The crystals were transferred into reservoir solution supplemented with 25% (vol/vol) glycerol in two steps for cryo-protection, then flash frozen by plunging into liquid nitrogen.

## Structure determination

X-ray diffraction data were collected at Brookhaven National Laboratory National Synchrotron Light Source (NSLS) beamline X29 and at Argonne National Laboratory Advanced Photon Source (APS) NE-CAT beamlines 24-ID-C and 24-ID-E. Data were integrated and scaled using HKL2000 (*Otwinowski and Minor, 1997*). The diffraction data were anisotropic. To compensate, isotropy was approximated by applying a positive b factor along a* and b* and a negative b factor along c* (*Table 1*), as implemented by the UCLA MBI Diffraction Anisotropy Server (http://services.mbi.ucla.edu/anisoscale/) (*Strong et al., 2006*), resulting in enhanced map features (*Figure 1C*, *Figure 1—figure supplements 1*, *2C,D*, *3D,E*, *Figure 3—figure supplement 1*, *Figure 4B*).

Initial electron density maps were calculated by molecular replacement using Phaser (*McCoy et al., 2007*) from a starting model of *Taq* Δ1.1σ$^A$-holoenzyme determined at 2.8 Å-resolution (unpublished). Two RNAP/DNA complexes were clearly identified in the asymmetric units. The models were first improved using rigid body refinement of each RNAP molecule and subsequently of 20 individual mobile domains using PHENIX (*Adams et al., 2010*). At this point, the electron density maps showed strong connected difference density for the nucleic acids, allowing unambiguous placement using COOT (*Emsley and Cowtan, 2004*). Detailed nucleic acid modeling was facilitated using available models of complexes with promoter fragments: σ$_4^A$/−35 element DNA complex at 2.4 Å (1KU7 [*Campbell et al., 2002*]), RNAP-holoenzyme/us-fork DNA at 6.5 Å-resolution (1L9Z [*Murakami et al., 2002b*]), σ$_2^A$/nt − strand −10 element DNA at 2.1 Å (3UGO [*Feklistov and Darst, 2011*]), RNAP-holoenzyme/downstream-fork DNA at 2.9 Å (4G7H [*Zhang et al., 2012*]), RNA/DNA hybrid at 2.5 Å (2O5I [*Vassylyev et al., 2007*]). The resulting models were improved using deformable elastic network (DEN) refinement (*Schröder et al., 2010*) with noncrystallographic symmetry (NCS) restraints using CNS 1.3 (*Brunger et al., 1998*) performed on the Structural Biology Grid portal (*O'Donovan et al., 2012*), followed by iterative cycles of manual building with COOT (*Emsley and Cowtan, 2004*) and refinement with PHENIX (*Adams et al., 2010*).

In the RPo structure, the ss t-strand DNA from −11 to −4 was only modeled in one complex of the asymmetric unit. In the other complex, strong, connected Fourier difference density for this segment of DNA was observed but the density was relatively featureless and we were unable to model this segment of the DNA. In the us-fork (−11 bp) complex, the t-strand T$_{−11}$ was modeled in only one complex of the asymmetric unit. In the other complex, density for this base was absent.

## Resolution limit and structure validation

We follow the criteria of *Karplus and Diederichs (2012)*, who showed that the $R_{merge}$ statistic commonly used to evaluate data quality is 'seriously flawed' and should not be used (*Diederichs and Karplus, 1997*), and that the commonly used criteria of $<I>/σI > 2$ also results in the loss of much useful crystallographic data (*Karplus and Diederichs, 2012*). *Karplus and Diederichs (2012)* showed, using objective and unbiased analyses, that inclusion of weak X-ray diffraction data ($R_{merge}$ values $>> 1.0$ and $<I>/σI << 1$) resulted in improved structural models. An improved statistic, CC* (essentially a Pearson correlation coefficient), was introduced that provides a single statistically valid guide for deciding whether diffraction data are useful.

Since most of the analyses described herein were performed from the RPo structure, we justify the inclusion of diffraction data to 4.14 Å-resolution for this case. Data in the highest resolution shell

(4.29–4.14 Å) are very weak when examined by standard criteria (high $R_{pim}$ values and $<I>/\sigma I = 0.8$, Table 1), but have good multiplicity (21.6) and completeness (99.8%), and yield a CC1/2 of 0.157, which is significantly different from zero for the large sample size (16,966 unique reflections) at exceedingly low p values (Rahman, 1968). That the highest resolution shells contain useful data and not noise is reflected in the observation that the $R_{free}$ and $R_{work}$ for the model refinement do not diverge (Figure 1—figure supplement 2, Figure 4—figure supplement 1). Inclusion of higher resolution data resulted in unacceptably low completeness in the highest shells due to the data anisotropy.

In the final $2F_o - F_c$ electron density maps, numerous protein side chains were resolved, including many that appeared to form important protein/nucleic acid interactions. To confirm these protein side chain positions, we produced unbiased difference Fourier maps using a simulated annealing omit procedure. Protein segments flanking the side chains in question were removed completely from the structural model, and the modified models were subjected to simulated annealing refinement using PHENIX (Adams et al., 2010). We used the following annealing temperatures (K), 1000; 2500; 5000; 10,000. All temperatures gave the same result (recovery of electron density for the omitted side chains), but the 5000 and 10,000 K refinements gave rise to obvious local structural distortions (expected for such high annealing temperatures with our low-resolution data) so the unbiased $2F_o - F_c$ maps were calculated from the 2500 K annealing refinements (Figures 2C,D, 3D,E).

## Kinetic measurements

### Preparation of Eco core RNAP, σ⁷⁰, and σ⁷⁰ mutants

Eco core RNAP was overexpressed and purified from Eco BL21 (DE3) cells co-transformed with pGEMABC (encoding Eco RNAP rpoA, rpoB, and rpoC; Addgene plasmid 45398) and pACYCDuet-1_Ec_rpoZ (encoding rpoZ) as described (Murakami, 2013). Eco σ⁷⁰ was overexpressed and purified as described previously (Feklistov and Darst, 2011). Eco σ⁷⁰ W433A and Y394A substitutions were generated by site-directed mutagenesis of pGEMD-σ⁷⁰ and purified using the same procedure as wild-type σ⁷⁰.

### Preparation of DNA for kinetic measurements

A 135 bp λ $P_R$ promoter with Cy3 label at position +2 of the nontemplate strand was prepared using a 79 nt long synthetic oligonucleotide containing amino-dT at +2:

ATCTATCACCGCAAGGGATAAATATCTAACACCGTGCGTGTTGACTATTTTACCTCTGGCGGTG ATAATGGTTGCA/iAmMC6T/GT

The oligonucleotide was modified with Cy3-NHS and purified by reverse phase HPLC. The duplex was then prepared by Taq DNA polymerase extension of a partial duplex formed by mixing 0.25 μM Cy3-labeled non-template strand and 0.275 μM 79 nt template strand (TGCTGACTGCTTAATCGCTTC TAGGGATATAGGTAATTCCATACCACCTCCTTACTACATGCAACCATTATCACCGCCA) containing at the 3′-end a 23 bp sequence complementary to the 3′-end of the nontemplate strand. Extended duplex was purified on a 1 ml Resource Q column (GE Healthcare Bio-Sciences, Marlborough, MA, United States) using a gradient of 0–1 M NaCl in 25 mM Tris–HCl (pH 8), 10 μM EDTA. Fractions containing labeled promoter were precipitated with ethanol to remove salt.

### Mechanistic model

Quantitative mechanistic studies have found at least two kinetically significant intermediates (designated $I_1$ and $I_2$) on the pathway to formation of RPo by Eco RNAP at the λ $P_R$ promoter (Davis et al., 2007; Gries et al., 2010; Saecker et al., 2011):

$$R + P \underset{\substack{\text{rapid} \\ \text{equilibrium}}}{\overset{k'_1}{\underset{k'_{-1}}{\rightleftarrows}}} I_1 \underset{\substack{\text{slow}}}{\overset{k'_2}{\underset{k'_{-2}}{\rightleftarrows}}} I_2 \underset{\substack{\text{rapid} \\ \text{equilibrium}}}{\overset{k'_3}{\underset{k'_{-3}}{\rightleftarrows}}} RPo, \qquad (1)$$

where the interconversion between $I_1$ and $I_2$ is rate-limiting in both directions (Buc and McClure, 1985; Saecker et al., 2002). The rate limiting step in the forward direction is the conversion of $I_1$ to $I_2$, so under

standard solution conditions, $I_2$ is never significantly populated (*Gries et al., 2010*). Because $I_2$ is not significantly populated under the conditions of association experiments, the three-step mechanism simplifies to the two-step mechanism (*Figure 5A*), where $I_1$ = RPi. Since the kinetics observed in the forward direction are well fit by a single exponential (*Figure 5B*), we deduce that RPi does not give rise to a significant fluorescence signal in our assay.

In the reverse direction, however, rapid destabilization of RPo (such as with 1.1 M NaCl used here) generates a burst of $I_2$ (*Kontur et al., 2006*; *Gries et al., 2010*). The complexity and shapes of the dissociation curves observed by our fluorescence assay are consistent with the detection of a transient burst of $I_2$ after challenging pre-formed RPo with 1.1 M NaCl (*Figure 5—figure supplement 1*) (*Gries et al., 2010*). Real-time observation of $I_2$ is an important finding that merits further, quantitative study but is beyond the scope of this study. Instead, we have characterized the overall dissociation rate ($k_{off}^{app}$) by fitting the dissociation curves with a single exponential, which reveals the gross (>fourfold) differences in overall dissociation rates observed between wild type and mutant σ's (*Figure 5D*, *Figure 5—figure supplement 1*).

## Forward kinetics

To measure the kinetics of RPo formation, *Eco* RNAP holoenzyme was loaded in one syringe of a stopped-flow instrument (SF-300X, KinTek Corporation, Austin, TX, United States) and Cy3-labelled promoter DNA in the other. After rapid mixing at the indicated temperature (37°C or 25°C), the final concentrations were: promoter DNA, 0.3 nM; RNAP, 2 to 150 nM in binding buffer (20 mM HEPES, pH 8.0, 100 mM K-Glutamate, 10 mM $MgCl_2$, 1 mM DTT). Cy3 fluorescence emission was measured in real time with a 586/20 single-band bandpass filter (Semrock) and excitation at 550 nm. The kinetics of Cy3 fluorescence were determined at various RNAP concentrations and fit to a single exponential equation (*Figure 5B*):

$$F_t = F_\infty + (F_0 - F_\infty)e^{-k_{obs}t}, \tag{2}$$

where $F_t$ is the fluorescence intensity of Cy3 as a function of time ($t$), $F_0$ is the initial fluorescence intensity, $F_\infty$ is the fluorescence intensity at $t = \infty$, and $k_{obs}$ is the pseudo-first-order observed rate constant of the increase in Cy3 fluorescence. The data were interpreted assuming the following kinetic scheme (*Figure 5A*; [*McClure, 1980*; *Buc and McClure, 1985*]):

$$R + P \underset{}{\overset{K_d}{\rightleftharpoons}} RPi \xrightarrow{k_2} RPo, \tag{3}$$

where the initial RNAP/promoter complex (RP$_i$) existing in rapid equilibrium with free promoter and RNAP (described by a dissociation equilibrium constant $K_d$) is converted in a rate-limiting step to RPo (described by the rate constant $k_2$). To obtain $K_d$ and $k_2$ the observed rate constants ($k_{obs}$, average values determined from >3 replicates) were plotted against RNAP concentrations (*Figure 5C*) and the data were fit to a hyperbolic equation (*Saecker et al., 2002*):

$$k_{obs} = \frac{k_2[\text{RNAP}]}{[\text{RNAP}] + K_d}. \tag{4}$$

## Reverse kinetics

Cy3-labeled DNA promoter fragments (0.3 nM) in binding buffer were mixed with RNAP-holoenzyme (100 nM) and incubated at 37°C for 20 min to preform RPo. They were rapidly mixed in the stopped-flow instument with the same buffer but resulting in a final NaCl concentration of 1.1 M. The kinetics of high-salt induced RPo decay was recorded in the same manner as for the forward direction. Averaged time traces from ≥3 replicates were fit to a single exponential *Equation 2* corresponding to a simplified kinetic scheme:

$$RPo \xrightarrow{k_{off}^{app}} R + P. \tag{5}$$

## Accession numbers

The structure factor files and X-ray crystallographic coordinates have been deposited in the Protein Data Bank under ID codes 4XLP (*Taq* holoenzyme/us-fork (−12 bp) complex), 4XLQ (*Taq* holoenzyme/us-fork (−11 bp) complex), and 4XLN (*Taq* RPo).

## Acknowledgements

We thank T Heyduk for assistance with fluorescent promoter synthesis and MT Record and R Saecker for helpful discussions and advice on the kinetic analysis, and EA Campbell for assistance with sequence alignments. We thank D Oren and The Rockefeller University Structural Biology Resource Center for technical assistance (supported by grant number 1S10RR027037 from the National Center for Research Resources of the NIH). We thank KR Rajashankar and F Murphy [APS Northeastern Collaborative Access Team (NE-CAT) beamlines] and W Shi (NSLS beamline X29) for support with synchrotron data collection. This work was based, in part, on research conducted at the APS and NSLS and supported by the U.S. Department of Energy, Office of Basic Energy Sciences. The NE-CAT beamlines at the APS are supported by Award RR-15301 from the NCRR at the NIH. Work at NSLS X29 was made possible by the Center for Synchrotron Biosciences grant, P30-EB-009998, from the National Institute of Biomedical Imaging and Bioengineering (NIBIB). BB was supported by a Merck Postdoctoral Fellowship (The Rockefeller University) and an NRSA (NIH F32 GM103170). This work was supported by NIH grants R01 GM038660 to RL and R01 GM053759 to SAD.

## Additional information

### Funding

| Funder | Grant reference | Author |
| --- | --- | --- |
| National Center for Research Resources (NCRR) | 1S10RR027037, RR-15301 | Seth A Darst |
| National Institute of Biomedical Imaging and Bioengineering (NIBIB) | P30-EB-009998 | Seth A Darst |
| The Rockefeller University | Merck Postdoctoral Fellowship | Brian Bae |
| National Institute of General Medical Sciences (NIGMS) | F32 GM103170 | Brian Bae |
| National Institute of General Medical Sciences (NIGMS) | R01 GM038660 | Robert Landick |
| National Institute of General Medical Sciences (NIGMS) | R01 GM053759 | Seth A Darst |

The funders had no role in study design, data collection and interpretation, or the decision to submit the work for publication.

### Author contributions

BB, AF, Conception and design, Acquisition of data, Analysis and interpretation of data; AL-N, Conception and design, Contributed unpublished essential data or reagents; RL, Conception and design, Analysis and interpretation of data, Drafting or revising the article; SAD, Conception and design, Acquisition of data, Analysis and interpretation of data, Drafting or revising the article

## Additional files

### Major datasets

The following datasets were generated:

| Author(s) | Year | Dataset title | Dataset ID and/or URL | Database, license, and accessibility information |
| --- | --- | --- | --- | --- |
| Bae B, Darst SA | 2015 | Crystal structure of T. aquaticus transcription initiation complex containing upstream fork promoter | http://www.rcsb.org/pdb/search/structidSearch.do?structureId=4XLP | Publicly available at the RCSB Protein Data Bank (Accession no: 4XLP). |
| Bae B, Darst SA | 2015 | Crystal structure of T. aquaticus transcription initiation complex containing upstream fork (−11 base-paired) promoter | http://www.rcsb.org/pdb/search/structidSearch.do?structureId=4XLQ | Publicly available at the RCSB Protein Data Bank (Accession no: 4XLQ). |

| Author(s) | Year | Dataset title | Dataset ID and/or URL | Database, license, and accessibility information |
|---|---|---|---|---|
| Bae B, Darst SA | 2015 | Crystal structure of T. aquaticus transcription initiation complex containing bubble promoter and RNA | http://www.rcsb.org/pdb/search/structidSearch.do?structureId=4XLN | Publicly available at the RCSB Protein Data Bank (Accession no: 4XLN). |

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
