## [Decision Letter]

Thank you for submitting your work entitled “Structure of a bacterial RNA polymerase holoenzyme open promoter complex” for peer review at *eLife*. Your submission has been favorably evaluated by Richard Losick (Senior Editor), and three reviewers, one of whom is a member of our Board of Reviewing Editors.

The following individuals responsible for the peer review of your submission have agreed to reveal their identity: Stephen Harrison (Reviewing Editor) and Carol Gross (peer reviewer).

The reviewers have discussed the reviews with one another and the Reviewing Editor has drafted this decision to help you prepare a revised submission.

The structure of the bacterial RNA polymerase with an intact open promoter, including a full transcription bubble, integrates data and conclusions from many partial structures. Although most (not all) the conclusions have been anticipated, at least in part, in analyzing the partial structures, the complete picture is essential both to be sure one is putting together the elements correctly and to show the significant of specific, conserved features – in particular the W-dyad. Despite a recent related paper from Steitz and coworkers, the manuscript adds to our understanding of the mechanisms of transcription initiation and is suitable for *eLife* after minor revision. Its impact is especially clear when taken with the sister paper describing the activation mechanism of CarD through buttressing the kinked DNA at the edge of the bubble.

Essential revisions:

1) Technical

a) The second paragraph of the subsection “Structural role of σ^A^ aromatic residues in forming and stabilizing the upstream ds/ss junction of the transcription bubble” states that Y217 appears positioned to stack on the -11(t) base. The authors should provide a distance between Y217 and the -11(t) base to clarify the picture.

b) At the end of the subsection “Structural role of σ^A^ aromatic residues in forming and stabilizing the upstream ds/ss junction of the transcription bubble”, the authors refer to the stacking interactions between Y217 and the -11(t) base. Density for Y217 is presented in Figure 4. As the model comes from the structure at lower resolution (4.6 Å), the authors should provide an omit map density figure for Y217, just as they did in cases of other residues they discuss.

c) Models presented in Figure 1 and Figure 6 contain a Mg^2+^ ion. Did the authors infer the position of the metal ion from other, higher resolution structures or is its density present in the structure?

d) The crystallographic statistics table suggests that the authors might be overestimating the resolution of the structure. They do carefully explain the reason behind using CC parameters instead of R factors to determine the cutoff, but CC_1/2_ is very low in the highest resolution shell and I/σI is quite small. A good criterion would be to set the resolution limit such that the highest resolution data still provide improvement in the model, with somewhat more convincing CC and I/σI statistics. Setting the resolution to slightly lower values would probably not alter the quality of the electron density maps, and this is of course what counts in the end. Or does refinement become unstable? The authors should consider these points and provide an updated table should they revise their resolution estimate (or provide direct evidence that the highest resolution shell is indeed contributing to the interpretability of maps).

e) When calculating the SA omit maps, the authors apparently just omitted side chains. If a short segment of chain had been omitted completely (main- and side-chain), would the SA omit maps have been similar (as one should expect)? The authors might want to re-do that bit of the computation, just to reassure us all.

2) Context of other work

a) Refer to recent paper by Zuo and Steitz in the Introduction and summarize its main conclusion.

b) It would be useful to have a brief discussion of the similarities and differences between the new structure and previous models. Such a discussion would be a tremendous service to the broad readership of this field, many lacking the knowledge to put this work in a coherent intellectual framework. As examples, what is the predicted impact of deleting region 1.1 from *TTh*σ^A^ and how does the structure relate to that of *Taq* (which retains region 1.1)? Are there any discernible differences between this structure and the higher-resolution structures that only contain the downstream edge of the bubble? What is the impact of lacking a 5′ triphosphate, especially in comparison with the [9] paper, which also examines the clash between the nascent RNA chain and σ_3.2_.

c) It is very good for a broader readership that the authors end with a comparison of their results to the eukaryotic system and a more conceptual discussion of how DNA opening is achieved in different transcriptions systems. We recommend that this paragraph refer to the proposed analogies and differences between σ and TFIIB (Kostrewa et al., Nature 2009 and Liu et al., Science 2010). In particular, note that TFIIB, like σ, reaches close to the upstream edge of the bubble but that TFIIB and σ elements occupying this position are different. One reviewer believes this is a major reason why DNA opening is different in bacterial and eukaryotic systems and that this is the principal point the authors wish to make with the concluding paragraph in the Discussion, but they do not explicitly state it. They should do so. In particular, the region in TFIIB that corresponds to the σ_3.2_ region extends only to the RNA nucleotide at the upstream position -6, not to -4, and the course of its polypeptide chain is reversed compared to σ.

d) If possible, provide a figure depicting residue conservation in the region discussed in the paper.

e) The authors should reconsider the use of the term ‘σ-finger’. It was introduced by analogy with the eukaryotic TFIIB element ‘β-finger’, a hairpin structure proposed in 2004 based on lower resolution X-ray data that could not be confirmed when higher resolution became available. The authors could change the name and still say (in parenthesis) something like (‘also referred to as σ-finger’). Using different terms for the σ and TFIIB elements occupying similar positions near the upstream edge of the bubble is appropriate and will make clearer the key point of the discussion – that the two mechanisms differ.

---

## [Author Response]

*Essential revisions*:

1) Technical2

*a) The second paragraph of the subsection “Structural role of σ*^*A*^
*aromatic residues in forming and stabilizing the upstream ds/ss junction of the transcription bubble” states that Y217 appears positioned to stack on the -11(t) base. The authors should provide a distance between Y217 and the -11(t) base to clarify the picture*.

We have changed the paragraph in question (now fourth paragraph of the subsection “Functional role of σA aromatic residues in forming and stabilizing the upstream ds/ss junction of the transcription bubble”) as well as Figure 3.

*b) At the end of the subsection “Structural role of σ*^*A*^
*aromatic residues in forming and stabilizing the upstream ds/ss junction of the transcription bubble”, the authors refer to the stacking interactions between Y217 and the -11(t) base. Density for Y217 is presented in*
Figure 4*. As the model comes from the structure at lower resolution (4.6 Å), the authors should provide an omit map density figure for Y217, just as they did in cases of other residues they discuss*.

Figure 4 has been revised to include the omit difference density.

*c) Models presented in*
Figure 1
*and*
Figure 6
*contain a Mg*^*2+*^
*ion. Did the authors infer the position of the metal ion from other, higher resolution structures or is its density present in the structure?*

The density for the Mg^2+^-ions is present in the maps. As an indication of this, the refined B- factors for the two Mg^2+^-ions are 136.4 Å^2^ and 142.08 Å^2^ (average of 139.24 Å^2^), while the average of the B-factors of ligand oxygens coordinating the Mg^2+^-ions (the RNA 3′-hydroxyls, as well as oxygen atoms from β′D739, 741, and 743) is 122.12 Å^2^ (not very different).

*d) The crystallographic statistics table suggests that the authors might be overestimating the resolution of the structure. They do carefully explain the reason behind using CC parameters instead of R factors to determine the cutoff, but CC*_*1/2*_
*is very low in the highest resolution shell and I/σI is quite small. A good criterion would be to set the resolution limit such that the highest resolution data still provide improvement in the model, with somewhat more convincing CC and I/σI statistics. Setting the resolution to slightly lower values would probably not alter the quality of the electron density maps, and this is of course what counts in the end. Or does refinement become unstable? The authors should consider these points and provide an updated table should they revise their resolution estimate (or provide direct evidence that the highest resolution shell is indeed contributing to the interpretability of maps)*.

For the main dataset used for analyses in this manuscript (the full RPo with the 4mer RNA), the I/σI falls off as follows near the highest resolution cutoff:I/σIResolution (Å)CC_1/2_24.770.591.54.490.471.04.260.280.54.140.090.334.000.059

We chose these five resolution cutoffs (4.77, 4.49, 4.26, 4.14, 4.00) to test the validity of the data using essentially the same procedure as described in Karplus and Diederichs (2012; Figure 1). For a starting model (the same starting model was used for each refinement), we used the model refined at 4 Å-resolution, but then set all the B-factors to 100 Å^2^ and randomly displaced all the atoms of the structure by a distance with a mean coordinate error of 0.5 Å using phenix.pdbtools (introducing severe distortions in the structure). We then used PHENIX to refine the structure at resolution X (say 4.77 Å), then at resolution Y (4.49 Å), then compared the R_free_ of both refinements, all calculated at resolution X (so direct R_free_ comparison is meaningful). If the R_free_ of the refinement at resolution Y (calculated at resolution X) is lower than the R_free_ of the refinement at resolution X (calculated at resolution X), then the additional higher resolution data between resolution X and Y is contributing meaningfully to the refinement and improving the overall structure. If the additional higher resolution data between resolution X and Y is essentially noise, then the R_free_ calculated in this manner would be expected to increase. To test the characteristics of the diffraction data more thoroughly, we performed the sets of refinements in two different ways. First, we performed the PHENIX refinements using a standard protocol the same as we used for the refinements described in the manuscript, with NCS and secondary structure restraints turned on. In a second set of refinements, we used the same protocol but with all restraints turned off.High res. cutoff (Å)I/σIR(res. X)R_free_(res. X)R(refined at res. Y but calculated at res. X)R_free_(refined at res. Y but calculated at res. X)R_free_(refined at res. Y but calculated at res. X) – R_free_(res. X)With NCS and secondary structure restraints4.77226.5530.074.491.527.1830.8625.7529.75-0.324.26128.2932.0727.0830.8-0.064.140.528.3232.0827.7831.75-0.324.40.3329.2532.8628.6532.310.23Without NCS and secondary structure restraints4.77226.2529.784.491.526.9130.6325.7529.5-0.284.26128.231.8827.0830.57-0.064.140.528.5732.1627.7831.84-0.044.40.3329.0832.7428.6532.15-0.01

The refinements with restraints (top set of rows) suggest that the data between 4.14 – 4 Å resolution are decreasing the quality of the structure, while the unrestrained refinement suggests this high resolution shell may contribute very weakly. Examination of the electron density maps calculated at 4.14 or 4 Å cutoff reveals very minor improvements: for instance, at 4.14 Å resolution the densities for σW256 and the T_-12_(nt) base (which are stacked on each other) are connected by a thin sliver of density, while at 4 Å resolution the densities are distinct. However, these ‘improvements’ in the electron density maps can hardly be said to increase the ‘interpretability’ of the maps. Therefore, we have revised the high resolution limit to be 4.14. The entire manuscript and the crystallographic table have been revised to reflect this.

*e) When calculating the SA omit maps the authors apparently just omitted side chains. If a short segment of chain had been omitted completely (main- and side-chain), would the SA omit maps have been similar (as one should expect)? The authors might want to re-do that bit of the computation, just to reassure us all*.

We have calculated SA omit maps where short segments of the chain (main- and side- chain) were omitted (surrounding the residues in question). The resulting SA omit difference maps (2Fo – Fc) are shown (Figures 2 and 3) instead of the difference maps from removing just the side chains.

2) Context of other work

*a) Refer to recent paper by Zuo and Steitz in the Introduction and summarize its main conclusion*.

The new Zuo and Steitz paper is introduced in the last paragraph of the Introduction and in the first paragraph of the Discussion. Results of Zuo and Steitz relevant to our study are also discussed in a new section of the Discussion (‘Transcript elongation, scrunching, and σ-release”).

*b) It would be useful to have a brief discussion of the similarities and differences between the new structure and previous models. Such a discussion would be a tremendous service to the broad readership of this field, many lacking the knowledge to put this work in a coherent intellectual framework. As examples, what is the predicted impact of deleting region 1.1 from* TTh*σ*^*A*^
*and how does the structure relate to that of* Taq *(which retains region 1.1)? Are there any discernible differences between this structure and the higher-resolution structures that only contain the downstream edge of the bubble? What is the impact of lacking a 5′ triphosphate, especially in comparison with the*
[9]
*paper, which also examines the clash between the nascent RNA chain and σ*_*3.2*_.

We have included a discussion of our new structure compared with previous models (generated from RNAP complexes with promoter fragments) in the first paragraph of the Discussion.

Region 1.1 of σ^A^, which is present in all primary σ's (including *Taq* and *Tth*), occupies the RNAP active site channel in the RNAP holoenzyme (Mekler et al., 2002; Bae et al., 2013) but is expelled from the channel upon the formation of RPo (Mekler et al., 2002; [95]). Accordingly, deletion of region 1.1 is known to affect the kinetics of RPo formation and dissociation in complex ways (Wilson and Dombroski, 1997; Vuthoori et al., 2001; Ruff et al., 2015), but region 1.1 is not expected to alter the protein/DNA interactions in the final RPo. Only one structure has managed to capture region 1.1 in the context of RNAP holoenzyme (our structure of region 1.1 trapped in the RNAP active site channel of Eco holo; Bae et al., 2013). Otherwise, at least in our experience, we had to either truncate region 1.1 in order to get crystals ([66], etc.), or if region 1.1 was present, it was disordered (many unpublished observations). Therefore, we chose to use Δ1.1σ^A^ for this study to increase the chances of crystallization. Zuo and Steitz crystallized Eco RPo with full-length σ^A^ (containing region 1.1) but it was not shown that region 1.1 was actually in the crystals (region 1.1 is linked to the rest of σ by a long linker and is therefore very susceptible to proteolytic degradation) – if it was still present, region 1.1 was not seen in the active site channel (since it was full of promoter DNA) and its location outside the channel was not revealed since it was disordered. Since region 1.1 was absent from our analysis (and it is not expected to alter RNAP holoenzyme/promoter interactions in the final RPo), we chose not to include a discussion regarding the presence/absence of region 1.1.

We have included a discussion of the effect of the presence/absence of the RNA 5′-triphosphate on the initial RNA transcript/σ_3.2_-loop interactions (in the subsection “Transcript elongation, scrunching, and σ-release” in the Discussion).

*c) It is very good for a broader readership that the authors end with a comparison of their results to the eukaryotic system and a more conceptual discussion of how DNA opening is achieved in different transcriptions systems. We recommend that this paragraph refer to the proposed analogies and differences between σ and TFIIB (Kostrewa et al., Nature 2009 and Liu et al., Science 2010). In particular, note that TFIIB, like σ, reaches close to the upstream edge of the bubble but that TFIIB and σ elements occupying this position are different. One reviewer believes this is a major reason why DNA opening is different in bacterial and eukaryotic systems and that this is the principal point the authors wish to make with the concluding paragraph in the Discussion, but they do not explicitly state it. They should do so. In particular, the region in TFIIB that corresponds to the σ*_*3.2*_
*region extends only to the RNA nucleotide at the upstream position -6, not to -4, and the course of its polypeptide chain is reversed compared to σ*.

We appreciate the reviewers' suggestion and have modified the manuscript to describe similarities and differences in σ and TFIIB contacts to the upstream end of the RNA:DNA hybrid. We note that these contacts likely affect events in early initiation and promoter escape in particular, whereas contacts of σ to the upstream DNA fork junction are likely to be among the most significant determinants of open complex formation. We do not currently have access to a structure revealing contacts of TFIIB to the upstream DNA fork junction in open complexes or to dsDNA prior to opening (none has been published), although contacts of the TFIIB reader residues R64 and D69 to template strand bases at -7 and -8 evident in available structures may contribute to open complex stabilization. Given this lack of information about upstream fork-junction contacts by TFIIB, we think it is premature to assign explanations for the major differences in DNA opening between bacteria and eukaryotes. Assigning such differences to differences in σ vs TFIIB contacts was not the major point we sought to make in the final paragraph. Rather, we sought to describe the global similarities and differences in the regulation of initiation between bacteria and eukaryotes.

*d) If possible, provide a figure depicting residue conservation in the region discussed in the paper*.

A large alignment of 1002 primary σ subunit sequences has been generated (included as Supplementary file 1), and a sub-alignment is depicted in Figure 1—figure supplement 3. Specific information on the conservation of RNAP β′ and σ^A^ residues is also tabulated in Supplementary files 2 and 3.

*e) The authors should reconsider the use of the term ‘σ-finger’. It was introduced by analogy with the eukaryotic TFIIB element ‘β-finger’, a hairpin structure proposed in 2004 based on lower resolution X-ray data that could not be confirmed when higher resolution became available. The authors could change the name and still say (in parenthesis) something like (‘also referred to as σ-finger’). Using different terms for the σ and TFIIB elements occupying similar positions near the upstream edge of the bubble is appropriate and will make clearer the key point of the discussion – that the two mechanisms differ*.

All being equal, we prefer ‘σ-finger’, but we also understand the reviewers' points. We have changed ‘σ-finger’ to ‘σ_3.2_-loop’ throughout the manuscript.